# Correlation between Defects and Technical Wear of Materials Used in Traditional Construction

**DOI:** 10.3390/ma14102482

**Published:** 2021-05-11

**Authors:** Jarosław Konior, Mariusz Rejment

**Affiliations:** Department of Building Engineering, Faculty of Civil Engineering, Wroclaw University of Science and Technology, 50-370 Wrocław, Poland; mariusz.rejment@pwr.edu.pl

**Keywords:** traditional construction, basic materials, technical wear’ defects, point biserial correlation

## Abstract

The degree of technical wear of old buildings, which are made of basic materials (cement, concrete, steel, timber, plaster, brick) using traditional technology, is expressed by the size and intensity of damage to their elements. The topic of the research concerns old residential buildings from the turn of the 19th and 20th centuries, which are located in the downtown district of Wroclaw, Poland. The descriptive analysis and the analysis of the definitions of defects that occur in the elements of residential buildings, which were performed as random analyzes, do not allow defects to be considered as measurable variables at a level of visual investigation. The major drawback of the method that is used by experts when assessing the technical condition of civil engineering buildings is that it does not numerically express the magnitude (strength) of the defects. Therefore, an attempt was made to numerically express the relationship (if such a relationship exists) between the occurred defects of buildings and the extent of their technical wear process. When calculating the strength of this relationship, the method of determining the point biserial correlation coefficient for the measurable property and the dichotomous property was used. It was found that the direction of the relation is right-hand for all the tested building elements, but the strength of the correlation between the detected defects and technical wear shows a considerable span and depends on the conditions of the apartment house’s maintenance. As a rule, damage caused by water penetration and moisture penetration always shows correlations of at least moderate strength.

## 1. Introduction

### 1.1. Justification for Undertaking the Research

A significant part of existing residential buildings in Poland are downtown tenement houses from the turn of the century—Figure 1. This period covers the last quarter (and even the middle) of the 19th century until the outbreak of World War I. The intensive development of urban housing was one of the consequences of the Industrial Revolution. In the territory of the former Prussian state, the conditions of peace that was concluded after the Battle of Sedan were an additional factor for this development. Housing construction at this stage of Wroclaw’s rapid development is therefore associated with the city-forming nature of the technical, economic, and social transformations of that period. The aim of the construction industry in post-war Poland was the quick and mass satisfying of minimal human demands. This led to the ensuring of a “roof over peoples’ heads” for those coming to cities. There are districts created in which buildings or parts of them already have a substandard level when compared to existing tenement houses. This is especially true for flats located on higher floors or in outbuildings. In addition, different building regulations and a different level of culture of residence apply in different countries. This dissimilarity affects, up to the present day, the quality of the housing resources in our country, which used to be divided into three annexations.

Therefore, certain parts of Wroclaw are dominated by buildings with a similar function, standard, and course of use, as well as similar construction methods. This fact indicates that it is advisable to combine activities indispensable for the renovation and modernization of individual houses with the comprehensive renovation of the city. This renovation, called revalorization, should be carried out in stages that include separated urban areas of buildings. Revalorization treatments require adaptation to the architectural shape and constructional-material solution of individual buildings that were erected using traditional methods and which are made from basic materials-cement, lime, aggregate, concrete, brick, wood, steel, and iron. This is associated with appropriate construction, material, and implementation solutions, whereas the stage that determines the correct application of these solutions involves the examination of the technical condition of groups of residential buildings, which are homogeneous in terms of their age and technology.

### 1.2. The Scope of the Research

It is expected that the shortage of housing in Poland will be compensated in two ways. The first involves the construction of new facilities. The second includes maintaining the existing resources by carrying out repairs and modernization. A gradual increase in residential investments to the level of 100,000 flats per year is expected. However, this only represents about 1% of the existing stock. At the same time, in buildings erected until 1918, there are about 770,000 apartments. This means that if the oldest buildings are not renovated and modernized, the total number of flats will start to decrease. The conclusions derived from this calculation should not unequivocally direct activities towards the rehabilitation of old buildings. The problem is complicated, and in this case a method of proceeding that involves the substantial deepening of the pre-investment stage, and the shortening of the previously well-prepared implementation stage, seems to be appropriate.

There are many reasons for modernizing old buildings. These include, among others, cultural factors, the limitation of demolition works, maintaining the safety of neighboring buildings, and the forecast of faster-than in the case of a new investment-renewal of urban routes. In turn, the quality and durability of the recovery activities are questionable. It is not always possible to use repair methods that are technically and economically effective. The global appraisal of the technical condition of objects, including the type and degree of destruction of individual elements of the structure, may contribute to the clarification of this problem. The assessment presented in this paper concerns 102 tenement houses in Wroclaw, which were erected mainly in the second half of the 19th century and at the beginning of the 20th century. Due to the size of the research group, the results of the assessment may be representative for buildings with similar architectural and construction features, and at the same time built according to similar legal principles. Due to the historical conditions of Poland, the western part of the country can be considered.

The data presented with regards to the entire country (or its western districts) can be applied to Wroclaw studies regarding this issue. The last “Population and Housing Census in Wroclaw”, which was carried out using the representative method in 2015, and which was prepared and published in 2016 [1], provides the following construction period and structure of ownership for flats in Wroclaw—Table 1.

Research concerning communal ownership has an advantage that—at least partially—there is an access to data that is collected in one place: Board of Communal Economy of Downtown District of Wroclaw. The data refers to the structure of housing resources with regards to their maintenance, and sometimes even to the technological cards of individual residential buildings (Regions of Residents Service). Therefore, with the assumption that the vast majority of residential buildings in the group to which the studied population belongs are municipal properties (incidents include cases where all apartments in the building were purchased by users and, consequently, the form of ownership was changed to a housing association or a housing cooperative), the number of municipal flats built before the outbreak of World War I is equal to approx. 45,000, i.e., 65% of all the municipal flats in Wroclaw—Table 1. Almost 1/4 of the inhabitants of Wroclaw live there. The greatest difficulty in the correct identification of the size of the studied population with regards to a separate group is the development and publication of official statistical data in relation to the user of the apartment, and not in relation to the inspected buildings. The author’s analysis of the structure of the selected sample in terms of the average value of the number of flats in randomly selected municipal residential buildings entitles 15 flats (with a variation index of 10%) per one considered tenement house to be adopted. Therefore, it can be assumed that there are approximately 3000 municipal buildings in use in Wroclaw of no less than 100 years old. The size of the isolated population of downtown tenement houses, which was estimated at 600, constitutes 20% of the group with the same form of ownership, a similar age and similar conditions of use and maintenance.

In the aforementioned comprehensive approach, the appraisal of the technical state of individual building elements is only a part of the issue. It should be noted that the entire issue is hierarchical in nature. At the highest level of the hierarchy there is the recognition of the previously specified goal and object of the assessment. This then allows for the specification of the urban and ecological criteria that refer to the group of existing housing (structured with regards to common quality features). The above criteria also affect the qualitative description of the assessment of the utility value, and together with this assessment enable requirements with regards to technical elements to be set—in the case of renovation and modernization of the building. The evaluation of the utility value also enables in the further steps (i.e., after the technical evaluation) the performance of calculations concerning the efficiency of renovation and modernization plans, i.e., the answer to the question: build or renovate? [2,3,4,5,6]. The example of the renovated tenement house is presented in the Figure 2.

Therefore, professionally conducted assessments of the technical condition of residential buildings are often enriched with a method of assessing the utility value and methodically conducted evaluation of their technical condition. The most representative are methods that are built in relation to specific local conditions (urban complexes, quarters of districts, identical urban routes), i.e., those that were built after identifying the factors that shape the residential microenvironment and the factors that influence the formulation of urban and revitalization criteria (which are particularly important in the case of residential buildings that were built using traditional methods). This identification results mainly from a series of works that were conducted by the academic experienced civil engineers under the supervision of Prof. *Kazimierz Czapliński* [7], and from the authors’ own works, for which studies [8,9] were used.

### 1.3. Literature Review

The literature review referred to sourced manuals and key publications, which characterize the approach, methods and models that are used for the technical and cost assessment of buildings, especially residential buildings. Table 2 lists the analyzed literature items with regards to the research topic and the applied research approach.

The social, cultural, and engineering diagnosis of the purposefulness of the revitalization or demolition of buildings is presented in an interesting way in papers [2,3,4,5,6]. A research attempt to assess the technical condition of downtown tenement houses in Wroclaw was described in detail in reports [7] and publications [8,9]. The approach to the investigation of building structures meets the broad requirements of their maintenance and use and is derived from standard BS 8210:2012 “Guide to facilities maintenance management” [10], with detailed research methods and tools being provided in publications [11,12,13,14,15,16]. The life cycle of a building structure is presented within the engineering approach in articles [17,18,19,20,21], and with regards to cost approach in papers [22,23,24,25,26,27]. Research methods and models with a full methodology of cause-effect relationships (“damage—technical wear”) were presented in publications [27,28], the analysis of which was supplemented with an approach to risk identification in the diagnosis of buildings [29,30]. In order to use statistical tools for the correlation of variables of various types, and for the extrapolation of the results of research to the general population, the authors’ knowledge from books and textbooks [31,32,33,34,35,36,37,38] was used.

## 2. Methodology and Methods

### 2.1. Subject of Research

The subject of the descriptive characteristics [7,8,9] includes a group of tenement houses located in a part of the downtown district of Wroclaw, Poland. The buildings are situated along the side streets (less often along the main streets). The urban layout of the downtown district has not been changed for years. The buildings are both front and outbuildings with modest architectural décor and low-budget functional standard. The tenement houses were built using traditional construction method, i.e., brick walls in longitudinal, usually three-bay, structural systems, solid floors. The 102 tenement houses were built mainly between the second half of the 19th century and the beginning of World War I.

The detailed technical characteristics of a selected set of downtown tenement houses (the group is compact in terms of functionality, and homogeneous with regards to material and structure) refer to the elements of the tenement houses and is synthetically presented in Table 3. The construction and material solutions of the described buildings are universal because they were erected using traditional methods from basic materials—cement, lime, aggregate, concrete, brick, wood, steel, and iron.

### 2.2. Research Sample

The research sample, which covered 102 technically evaluated apartment buildings from the downtown district of Wroclaw, was separated from a group of 160 tested buildings [7,8,9]. The dominant criterion in selecting the research sample was obtaining a comparable group of objects. Mutual comparability of the downtown tenement houses meant:compactness of the building development in the urban layout that has not been changed for years;a similar location along downtown side streets with an urban (but not representative style of a building façade) character;age coherence, i.e., a similar date of construction, maintenance and use in relation to historical and social aspects;construction and material similarity, especially in relation to the load-bearing structure of buildings;the same functional solutions, understood as the standard of apartment facilities and furnishings (in those days), and a defined standard of living for residents.

The method of selecting the research sample (on a more detailed level) was based on mutual similarity of all the technical solutions of the downtown tenement houses.

The selected research sample, according to the group of criteria listed above, is a representative sample in relation to the model of representativeness that is particular for the adopted aim of the research [31,32]. It contains all the values of the variables that could be recreated from the research carried out earlier by the authors using a different objective function than the one adopted in the research. Next, these values and variables were compiled and processed in such a way that it was possible to draw some conclusions about the cause-and-effect relationships among them in the general population. Because of the fact that the population structure and its characteristics were well recognized before, such a selection of the research sample can also be viewed as a conscious selection.

### 2.3. Technical Assessment

The obtained results of the measurements of the degree of technical wear of the elements of the inspected buildings are presented statistically in the Figure 3. The technical wear of horizontal and vertical anti-moisture insulations (reaching almost 100%) can be considered to be insignificant with regards to the amount of wear of the entire building due to their small (1%) share in the total cost. From the group of 10 elements that were selected for further analysis, the facades Z20 are worn to the highest degree (56.77%), and solid floors above basements Z4—to the lowest degree (43.16%). The dispersion of the outcomes is confirmed by the high values of the standard deviation and the coefficient of variation—the largest for facades Z20 (24.66% and 43%, respectively), and the smallest for foundations Z2 (8.83% and 20%, respectively). Other statistical quantities, such as the average value of deviations from the mean, variance and median, also reach their maximum and minimum values in the assessment of the same elements (Table 4). In the particular cases of the facades, the presented findings are confirmed by the description of the destruction, damage and various maintenance conditions, but the results concerning the foundations—due to the difficulty of the macroscopic evaluation—should not be considered as reliable. The amount of technical wear of the building, as a whole, was calculated when taking into account the share of the individual elements in its total cubature. Attention is drawn to the fact that two basic structural elements (structural walls aboveground Z7 and inter-story floors Z8) impact the total reconstruction cost of the building by 1/3. For a building with an average degree of technical wear equal to 48% and a different share of Z1–Z23 elements, the values of the standard deviation and the coefficient of variation are 10.94% and 23%, respectively—Figure 3.

The supplementation of the measurement results obtained by the team of experts with information, the scope of which was presented in the previous section of the paper, enabled intermediate conclusions (mainly concerning the technical state of the downtown tenement houses—the frequency of their damage and destruction, the scope of repairs, replacement, and reinforcement) to be presented. Some of them, formulated on a sample of 102 buildings, are described below:scratches or cracks in structural walls were noticed in 10 buildings;corrosion of steel beams of the floors above basements was found in 59 buildings, and in 28 cases it was surface corrosion;biological infestation of roof truss elements concerned 59 buildings, of which in six buildings the general condition of the truss was poor. Moreover, in 16 buildings, there was dampness of the roof structure due to leakage in the roofing;biological infestation of the wooden floors between stories occurred in 32 buildings, and it mostly affected the most endangered parts of the floors; extensive and advanced destruction was observed in 13 buildings;the dampness of underground walls was observed in 42 buildings, and of aboveground walls in 36 tenement houses. In five buildings, in the years 1983–1987, drying using electroosmosis, and a protection using electro injection, was performed; in two cases the treatments turned out to be ineffective;

When considering the issue with regards to the decision on the purposefulness of the renovation and modernization of the buildings, the condition of their structural elements was analyzed in its entirety. Five basic elements were distinguished, i.e., underground structural walls Z3, structural walls aboveground Z7, solid floors above basements Z4, inter-story wooden floors Z8, and the roof structure Z10. In the group of 102 assessed objects, there were 9 buildings in which significant destruction of all the distinguished elements was observed, and six buildings in which the destruction concerned four out of five of the analyzed elements. It can therefore be concluded that the renovation of 15 objects in the group of 102 buildings can be seen as ineffective.

### 2.4. Damage Characteristics

The theoretical grounds regarding intensity of formation of defects in residential buildings and their reliability has already been presented in [9]. The problem of the damage and destruction of residential buildings was described in detail in the literature, e.g., [10,13,15]—also in relation to the life cycle of a building [17,19] and the cost approach [22,25]. It can be found a classification of damage that is specific for different types of buildings, e.g., those built using industrialized construction methods or traditional construction methods. Unfortunately, the problem is much less recognized when more details are considered, i.e., when it concerns particular elements of traditional residential buildings. Therefore, the individual approach to the problem, described by the authors in this paper, might be interesting because of the fact that it connects the type and frequency of damage in the elements of downtown tenement houses with the current process of their technical wear. This aspect of the mutual relation, “damage-technical wear of building elements”, (i.e., the maintenance conditions and methods of using residential buildings in the downtown district), is presented in detail.

The result of detailed analysis of the technical reports [7] of the tested group of 102 downtown tenement houses included the identification of all the particular damage to their building elements at the basic level. The synthesis of the type of particular damage in relation to the same source of their formation (B, D), and the similarity of the consequences caused by them (A, B, C), made possible the damage to be semantically and generically arranged into the following four groups, i.e., A, B, C, D—see Table 5:mechanical damage to the structure and texture of building elements:
losses of brick and mortar in the solid floors above basements, underground walls, structural walls aboveground and stairs; losses in roofing and flashings;peeling off of the paint coatings on door joinery and external and internal plasters;decay of the brick, mortar and plaster of solid floors above basements, basement walls and walls aboveground; and also decay of external and internal plasters;cracks in the brick and plaster of solid floors above basements, basement walls and walls aboveground; cracks of external and internal plasters;peeling off of the paint coating of external and internal plasters;loosening of the plasters of inter-story ceilings, and also external and internal plasters;scratching of walls, plasters on inter-story floors, walls aboveground, and external and internal plasters;leakage of the roofing, water and sewage installations, gas installations, and window and door joinery;cracks in flashings, water and sewage installations, and gas installations;mechanical damage to stairs, flashings, all installations, window and door joinery, and external and internal plasters;falling off of external and internal plasters.damage caused by water and moisture penetration:
weeping of solid floors above basements, inter-story floors, underground walls, walls aboveground, stairs, the roof structure, roofing, window joinery, and external and internal plasters;corrosion of the brick in the underground walls and walls aboveground, flashings, water and sewage installations, and gas installations;dampness of solid floors above basements, inter-story floors, underground walls, walls aboveground, the roof structure, roofings, window joinery, and external and internal plasters;mold on the basement walls, walls aboveground, window joinery, and external and internal plasters;fungus on solid floors above basements, roofing, and external and internal plasters;surface corrosion of steel beams, solid floors above basements, and stairs;corrosion raid on the steel beams of solid floors above basements, and also on stairs;pitting corrosion of the steel beams of solid floors above basements, and also on stairs;decay of the wooden roof structure;flooding of solid floors above basements.damage revealed by the loss of the primary shape of wooden elements:
deformation of the wooden beams of inter-story floors;skewing of window and door joinery;warping of window and door joinery;sensitivity of the wooden beams of inter-story floors to the dynamic action of human weight;delamination of the wooden elements of the roof structure.damage to timber elements attacked by biological pest:
biological infestation of the wooden beams of inter-story floors, the roof structure, and window and door joinery;decay of the elements of wooden stairs, the roof structure, and window and door joinery.

The detailed analysis of the characteristics of the damage to elements of downtown tenement houses takes into account its four-level classification at the level of general synthesis. Moreover, the analysis separates generically integrated elementary damage, and due to this the number of types of damage increases to 30. It also includes a very important assumption, according to which the fact of the occurrence of the identified damage is described by number 1, and its absence by 0. These diagnostics assume that the group of experts only identified damage to elements of the analyzed buildings that are independent to their natural wear (those that occurred significantly), and described them with clear and decisive qualitative features as [7,8]: “significant”, “very significant”, “strong”, “total”, etc.

In the detailed characteristics of damage, it was possible to assign each elementary damage to its element, which was described by the degree of its technical wear. Therefore, the first step of the transition from a qualitative to quantitative notation in the construction of the cause-effect model “damage—technical wear” was made. The number of tenement houses subjected to technical assessment, which were maintained in satisfactory, average and poor conditions, enabled the frequency (probability) p(d)II, p(d)III, p(d)IV of the damage occurring in these states to be calculated. The extent of the research made possible to determine the total probability of the occurrence of 30 elementary types of damage—Table 5. All the results of the author’s own research, which were presented in the detailed characteristics, focused on the 10 building elements with the highest and most important share in the total of all building structures.

The detailed description of the damage was developed by examining the relationship between the intensity of its occurrence and the amount of technical wear of the selected elements of the tenement houses.

### 2.5. Research Model

The general scheme of the cause-and-effect relationship “damage—technical wear of building elements” results from a synthesis of the results of visual research of the selected sample of the tenement houses from the downtown district in Wroclaw [28]. Whenever the estimation of the technical condition of any chosen group of residential buildings and their building elements, the group of experts works at the intermediate stage of the proposed model, i.e., the analysis of the symptoms of the observed technical states. It cannot measure the causes (i.e., factors), but it can take their effect into account when conducting the estimation. The most important factors, that cause the accelerated destruction of downtown tenement houses, are described in the works [8,9]. In the case of effects (i.e., consequences), short-term effects (e.g., loss of utility value) and intentional effects (concerning decisions about the future of service life of a residential building) can be distinguished. Further actions depend on the adoption of one of the multi-criteria decision-making methods, e.g., according to [8]. The more reliably the research concerning the symptoms of damage to residential buildings’ elements is carried out in the observed states, the more reliable the premises for further decision-making analysis are.

#### 2.5.1. “Damage—Technical Wear” Correlation

The visual stage of determining the size of the symptoms of damage to the elements of downtown tenement houses consisted of identifying two types of variables:immeasurable—qualitative variables, i.e., single damage d_ij_;measurable—quantitative variables, i.e., the size of the degree of the technical wear of individual elements w_i_.

The descriptive and conceptual characteristics of the damage to the elements of tenement houses with regards to system analysis do not allow them to be accepted as measurable variables at this stage of work. The greatest disadvantage of the method used by experts to assess the technical condition of downtown tenement houses was that it did not quantify the size (strength) of the damage. It was not possible, even with the most detailed recreation of the technical documentation supported by verifying investigations, to distinguish the measurable value of e.g., “significant corrosion” of steel beams in stairs from “strong corrosion” of this element, and e.g., “significant wear” of the electrical installation from “significant wear” of a different element. With such a significant conceptual failure to specify the size of the elementary damage d_ij_, it was decided to determine the occurrence (or not) of damage in the binary system, in which {d_j_} = [0, 1]. Therefore, it was assumed that the damage to a building element—identified at the basic level—is a variable of a dichotomous nature.

After determining the type of variables w_i_ and d_ij_, an attempt was made to numerically express the correlation (if any) between them, i.e., to measure the impact of the occurred damage to the elements of the analyzed buildings on the size of the process of their technical wear. In order to calculate the strength of this relationship, the authors applied the method that determines the point biserial correlation coefficient for the measurable property w_i_ and for the dichotomous property d_ij_, which has been generally denoted as r(W). It is one of the few cases in the statistics of correlating properties of different types [32,37]. The correlation coefficient changes its value within the range [−1.1]. In damage sets D, for each type of elementary damage d_ij_ = d_i_ (when j = 1,2, …, m), and technical wear W, the following were determined:d_0_—the number of observations of variable d_i_ denoted with 0;d_1_—the number of observations of variable d_i_ denoted with 1;d_i_—a dichotomous variable that takes values 0 (d_i0_) or 1 (d_i1_); i = 1,2, …, n;

of course, d = d_0_ + d_1_ (if d is understood as the total number of all observations d_i_), and:w_i0_—value of the feature w_i_ for those units “i” for which the feature d_i0_ appears;w_i1_—the value of feature w_i_ for those units “i” for which the feature d_i1_ appears.w_i_—measurable variable; the values of this variable were split into two groups depending on whether d_j_ takes the values 0 or 1; i = 1,2, …, n;

The arithmetic means were then calculated in both groups:(1)$$\bar{w_{0}}=\frac{1}{d_{0}}\overset{d_{0}}{\underset{i=1}{\sum }}w_{i0}$$
(2)$$\bar{w_{1}}=\frac{1}{d_{1}}\overset{d_{1}}{\underset{i=1}{\sum }}w_{i1}$$
standard deviation (calculated for correlation r(W) using a differently defined relationship):(3)$$d\left(W\right)=\sqrt{\frac{d\sum_{i=1}^{d}w_{i}^{2}-{\left(\sum_{i=1}^{d}w_{i}\right)}^{2}}{d\left(d-1\right)}}$$
and as a result, on the basis of Equations (1)–(3), the point biserial correlation coefficient r(W) was determined as follows:(4)$$r\left(W\right)=\frac{\bar{w_{1}}-\bar{w_{0}}}{d\left(W\right)}\sqrt{\frac{d_{1}d_{0}}{d\left(d-1\right)}}$$

The presented method of associating defects of the elements of the analyzed buildings with their technical wear, which enables the direction and strength of this relationship to be determined, was used to investigate the impact of the damage on the occurrence of the process of technical wear in the following research states of the residential buildings:the observed state, where the correlation coefficient was determined in all five classes of the technical wear of building elements r(W) = r(We), and also additionally in each of the three middle maintenance states of the tenement houses r(We)II, r(We)III, r(We)IV;the theoretical state, in which the correlation coefficient was calculated for the technical wear, which was calculated in classes I-V according to time formulas and the two-variant assumption of durability T of building elements:
∗literature durability T* = T [8], r(W) = r(Wt*);∗maximum durability T** = t_max_, r(W) = r(Wt**).

The values of the calculated point biserial correlation coefficient for the element with the highest share in a building (construction walls of the overground W7) are given in the characteristics of its damage, which is described by the theoretical and observed states—Table 6.

The numerical image of the cause-effect relationship “damage—technical wear”, presented in the Table 5, was complemented by the calculated differences of the mean value “DE” of these values of the technical wear We, Wt* and Wt** for which d_i_ = 0 and d_i_ = 1. In this table, which contains the results of the author’s own research concerning the theoretical and observed states, the probability values p(d)II, p(d)III, p(d)IV of the occurrence of damage of the analyzed elements in the entire research sample were included as well.

#### 2.5.2. The Significance of the Correlation

The analysis of the cause-effect relationships (“damage—technical wear”), compiled simultaneously for the 10 selected building elements, indicates a significant range of the strength of these compounds within the same type of elementary damage d1–d30. Due to comparison the range of changes in the correlation of damage and technical wear with the direction of change of the ranges of partial probability p(d)II, p(d)III, p(d)IV, they were summarized in the Table 6. In this table, the association values of these damages, which correlate the most with the technical wear, i.e., r(W) > 0.5, were presented.

The observation of such a significant spread of the complicity the measurable variable w_i_ with the dichotomous variable d_i_ prompted the examination of the significance of this correlation in a sample with a size N ranging from 95 to 102 surveys of the 10 selected elements of the downtown residential buildings. The study of the significance of the correlation coefficient r(W) was carried out, as is the case with the *Pearson and Spearman* tests [8,36], using the Student’s t-statistic, which is formulated as follows:(5)$$t=r\left(W\right)\sqrt{\frac{d-2}{1-{\left[r\left(W\right)\right]}^{2}}}$$

With the number of the degrees of freedom of df = N−2. As a result, the exact probability p(r) of obtaining such a t-statistic value as the one obtained from the representative sample was calculated. It was assumed that there is the null hypothesis H_0_ (r(W) = 0) against both the alternative hypothesis H_1_(r(W) ≠ 0) and the determination of the two-sided area of the criterion. The probability p(r) corresponds to the observed significance level. In principle, in the case of associating properties of different types, it would not be a mistake to adopt a significance level of 10%. However, in order to clearly distinguish the types of damage that most strongly determine the degree of technical wear of building elements, it was assumed that if p(r) < 0.05, then the tested correlation is significant, and if 0.05 ≤ p(r) < 0.10, it can be concluded that there is a tendency to the sought relationship. Table 7 presents the values of the point biserial correlation coefficient r(W). The values that correlate most strongly (at the significance level of 5%), and the values that show a tendency of the occurrence of the association between the damage and the amount of technical wear of the components of the analyzed residential buildings, were highlighted.

#### 2.5.3. Extrapolation of the Correlative Results

All research samples of the 10 selected elements of the downtown tenement houses are statistically numerous samples (d > 30), and what is more they account for 15.8% to 17.0% of the general population of 600 buildings. In order to extrapolate the results of such a sample to the whole population and to determine the confidence intervals for the point biserial correlation coefficient in the general population, the authors applied the approximation of the normal distribution N (0.1). It was founded that each of the confidence intervals would cover the real value of r, (which is an estimation of the correlation coefficient in the population calculated from the sample) with the probability of 1 − p(r) = 0.95. For the adopted value of the distribution function Φ(x) = 0,95 in the normal distribution (with the mean value of 0 and standard deviation 1), the statistical value was read, and was in each case equal to x = 1.96. The correlation coefficient g(W) in the general population is defined by the lower and upper limits of the confidence interval in the relationship [32]:(6)$$r{\left(W\right)}_{d}<g\left(W\right)<r{\left(W\right)}_{g}$$
and therefore:(7)$$r\left(W\right)-x\frac{1-{\left[r\left(W\right)\right]}^{2}}{\sqrt{d}}<g\left(W\right)<r\left(W\right)+x\frac{1-{\left[r\left(W\right)\right]}^{2}}{\sqrt{d}}$$

It was founded as well that the square of the r(W) estimator corresponds to the percentage of the general population for which the obtained data can be referred to the confidence level of 95%. The confidence intervals of coefficients between the damages to the buildings’ elements and their technical wear in the general population, as well as their size, were determined. Those in which g(W) is at least of a moderate strength (r(W)_d_ > 0.45) were also distinguished as the least significant:g(W) < r(W)d = 0.45—correlation coefficient between “d” and “w”, which is weak in the population0.45 = r(W)d < g(W) < r(W)g = 0.70—correlation coefficient between “d” and “w”, which is of moderate strength in the population0.60 = r(W)d < g(W) < r(W)g = 0.80—correlation coefficient between “d” and “w”, which is quite strong in the population

## 3. Results

The results of the investigated cause-and-effect relationship (“damage—technical wear”) in the representative sample of downtown tenement houses built using traditional construction methods allow for the composing the following conclusions—Table 5, Table 6 and Table 7:predominantly, damage caused by water and moisture penetration (group II) has the greatest impact on the amount of technical wear of the elements of the analyzed tenement houses—0.54 on average, and this correlation is always significant;mostly, the correlation of at least moderate strength is shown by damage caused by water and moisture penetration (group II); only in the case of internal plasters and facades, can single mechanical damage to their structure and texture be considered moderate and quite strong (group I);damage manifested by the loss of the primary shape of wooden elements (group III) can be considered as insignificant; the exception is the skewing of window joinery with a correlation of 0.42, for which this defect determines a significant decline in the value of its utility function;the direction of the relationship is right-hand (positive) for all the 10 tested building elements, but the strength of the correlation between the occurring damage and its technical wear shows a wide range (from 0.00 to 0.84);the technical condition of each of the tested elements also shows the impact of the damage, which is characteristic for the design and material solutions of such an element, for example:
○damage to wooden parts of elements (ceiling beams, stair treads, roof trusses, window joinery) attacked by biological pests (group IV) r(W) ≅ 0.42;○mechanical damage to the structure and texture (group I), the significance of which applies only to those elements in which this defect may be the cause of the intensification of the impact of cumulative damage, e.g., basement and aboveground structural walls, as well as internal and external plasters (apart from foundations and solid floors above basements).extrapolation of the results of the investigated cause-effect relationship (“damage—technical wear”) in the representative sample of downtown residential buildings to the general population of 600 tenement houses leads to the following additional observations:
○for the adopted confidence level of 95%, moderate strength relationships can be related to 34–48% of the general population, and quite strong correlations—up to 49–71%;○in each of the tested elements (apart from wooden inter-story floors) there is at least one correlation coefficient g(W), which is determined in the general population; a moderate strength of the considered relationship (0.45 = r(W)_d_ < g(W) < r(W)_g_ = 0.70), or a fairly strong relationship (0.60 = r(W)_d_ < g(W) < r(W)_g_ = 0.80) was identified; a very strong correlation (g(W) > r(W)_d_ = 0.80) was not found.

## 4. Discussion and Conclusions

The article presents synthetic and analytical works, as well as model solutions concerning the problems of the technical maintenance and wear of residential buildings with a traditional construction. The authors have determined the cause-and-effect relationships between the occurrence of damage to the buildings’ elements (which is treated as an expression of their maintenance conditions) and the size of the technical wear of these elements. It has been done on a representative and purposefully deliberately chosen sample of 102 residential buildings constructed mainly between in the second half of the 19th century and at the beginning of the 20th century in the downtown district of Wroclaw, Poland.

The reliability and novelty of the research has been proved by new approach to unique correlations of two different kinds random and was published in the papers applying probabilistic and correlation methodology. The correctness of the test results for a representative group of old downtown apartment houses with traditional structure can be therefore summed up by the following conclusions:age of elements of old apartment houses with a traditional structure:
○is of secondary significance in the process of the intensity of loss of its useful values;○is not the essential size determining the course of their technical wear and tear;the technical extent of wear and tear of the components of an old residential building is determined by the conditions for its maintenance and use;the analysis of quantitative damage carried out by experiential methods of appraising the technical condition of the building shall indicate the nature and magnitude of the damage to its components which are characteristic of the relevant maintenance conditions;a study analysis of the processes of operation of residential objects and the basic dependencies of reliability theory made in it indicates that for the useful life of an object in which the working time to damage has an exponential distribution (this is in principle the life expectancy corresponding to the length of service of the dwellings concerned), the average remaining time of unsafe operation is constant at all times. Therefore, tenement houses, after some time of trouble-free operation, perform their use as new. The age of the elements of an old house is then of secondary importance in the process of the intensity of loss of its useful value;if assumed that the measure of matching the nonlinear mathematical models tested in the nonlinear regression method, as a function of the technical consumption of building elements over time, is the determination factor, then no more than 30% of the destruction of the elements is justified by the flow of time; age is therefore not a determinant of the technological consumption of the elements of the buildings analyzed.

The research works led to numerical expression the relationship between the occurred defects of buildings and the extent of their technical wear process. When calculating the strength of this relationship, the method of determining the point biserial correlation coefficient for the measurable property and the dichotomous property was used. This is one of the very few cases in statistics when properties of various types may be correlated. A number of works by *Nowogońska* [13,14] were studied in the methodical approach to the technical assessment of tenement houses but none of them concentrated on cause (damage)—effect (maintenance) model. The fuzzy approach presented in the publications of *Plebankiewicz*, *Wieczorek*, and *Zima* [22,23,24,25] was used in the assessment of the whole service life of a building objects but considered the cost aspects in their life cycle. The social, cultural and engineering diagnosis of the purposefulness of the revitalization or demolition of buildings is presented in an interesting way in papers by *Noonan*, *Power*, *Ástmarsson*, et al. [2,3,4,5,6]. All of them are mainly descriptive and do not present the results of damage measurements. The life cycle of a building structure is presented within the engineering approach in articles of *Chen*, *Frangopol*, *Silva*, *Grant*, *Saleh* [17,18,19,20,21] but all of them are representing similar approach to *Plebankiewicz* with regards to cost approach in papers [22,23,24,25]. Research methods and models with a full methodology of cause-effect relationships (“damage—technical wear”) were presented in publications of *Konior* and *Kapliński* [8,9,28], where methods and models for associating damage and technical wear were presented and discussed. Valuable works of *Lee* and *Terentyev* have also been analyzed and taken into account as a supplement with an approach to risk identification in the diagnosis of buildings [29,30].

It should be added that the rich research material, which is a result of several years of work by a team of experts and the author, has been arranged in a logical way, and constitutes one coherent study that is prepared for further processing. Therefore, an enormous emphasis in the paper was placed on the methodological side of the research (Section 2.4) and the purposeful and representative selection of the research sample (Section 2.2).

It should also be remembered that the new EU’s legislation will definitely affect our future research works in terms of energy performance of buildings to be renovated. Therefore, we point out that that the new EU Energy Efficiency Directive EU/2018/844 of 30 May 2018 [39] (amending Directive 2010/31/EU on the energy performance of buildings and Directive 2012/27/EU on energy efficiency) introduces a number of changes, new requirements and simplifications to some of the current rules mainly on energy consumption and sets out requirements for the renovation of existing buildings. According to the directive, EU members must develop long-term strategies for the renovation of existing residential and non-residential, public and private buildings. The aim of this strategy is to target multi-annual actions on the renovation of existing buildings. The renovation strategy will require the introduction of new legal and technical regulations as well as new systems to support the implementation of renovations. According to the authors of the article, the data and results of their research work can help the Polish government to introduce new technical requirements for the renovation of existing multi-family residential buildings in Poland.

## Figures and Tables

**Figure 1 materials-14-02482-f001:**
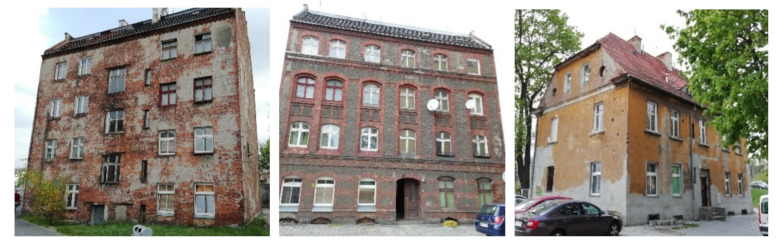
Selected examples of the analyzed tenement houses.

**Figure 2 materials-14-02482-f002:**
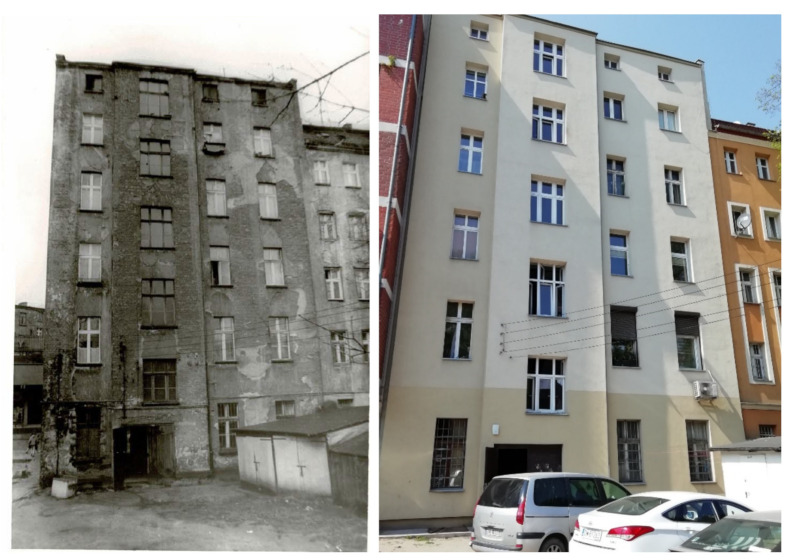
The tenement houses in Jedności Narodowej No 59 street before and after major renovation.

**Figure 3 materials-14-02482-f003:**
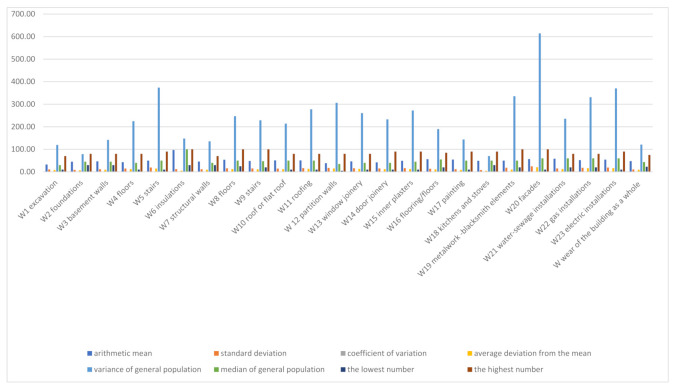
Basic statistical parameters of the technical wear of 23 building elements.

**Table 1 materials-14-02482-t001:** Flats in Wroclaw according to their construction period.

Subject of the Census	Total	In Buildings Constructed in the Years				
		Before 1918	1918–1944	1945–1970	1971–1995	1996–2010
Number of flats	204,293	46,139	42,656	36,163	41,882	37,453
Floor area [m^2^]	12,242,487	2,672,923	2,688,962	1,666,680	2,611,562	2,602,360
Population	643,782	140,137	134,805	88,236	131,021	124,270

**Table 2 materials-14-02482-t002:** Digest of references broken down to topics of research.

Ref.	Authors	Period	Topic of Research	Type of Approach
[2,3,4,5,6]	*Noonan, Power*, *Ástmarsson*	2008–2020	Social & technical aspects of housing	Decision making algorithms
[7,8,9]	*Czapliński, Konior*	1996–2020	Research sampling	Probability
[10]	*BS 8210:2012*	2012	Facilities maintenance management	N/A
[11,12,13,14,15,16]	*Macarulla*, *Antonini*, *Nowogońska*	2015–2020	Technical assessment of buildings	Stochastic & heuristic networks
[17,18,19,20,21]	*Chen, Frangopol*, *Silva*, *Grant*, *Saleh*	2008–2017	Lifecycle and service time of buildings	Systems analysis
[22,23,24,25,26,27]	*Plebankiewicz*, *Saleh*, *Noshadravan*	2008–2019	Life-cycle cost of buildings	Financial appraisal
[8,9,28]	*Konior*, *Kapliński*	2007–2021	Methods & models for associating damage & technical wear	Non-linear regression & fuzzy membership function
[29,30]	*Lee*, *Terentyev*	2017–2018	Risk identification of defects & damage occurrence	Risk management & mitigation
[31,32,33,34,35,36,37,38]	*McClave*, *Witte*, *Wasserman*, *at al.*	2001–2019	Statistical analysis & stochastic functions	Statistics & correlation

**Table 3 materials-14-02482-t003:** Descriptive characteristics of the technical assessment of the tested building elements—a selected fragment.

				Housing Development						Floors		Structural Walls			Roof		
	No. of Expertise Report	Location	Year of Construction	Compact	Semi-Compact	Free-Standing	Number of Stories	Basement *	Attic **	Above Basements	Inter-Story	Basement	Above Ground	Stairs	Construction	Roofing	Flashing
No	a		b	c	d	e	f	g	h	i	j	k	l	m	n	o	p
⋮	⋮	⋮	⋮	⋮	⋮	⋮	⋮	⋮	⋮	⋮	⋮	⋮	⋮	⋮	⋮	⋮	⋮
										solid	wooden	brick	brick	brick-steel	mono-pitched	asphalt roofing	galvanized sheet
78	21/90	Łowiecka 3	1885		#		5	C	S	extensive	surface	drying using	dampness	corrosion	complete	leakage	damaged
										corrosion	biological corrosion	electroosmosis	corrosion	damage	insect infestation	dampness	losses
										solid	wooden	brick	brick	wooden	purlin-column	roof tiles	galvanized sheet
79	22/90	Kilińskiego 25	1860	#			4	C	S	average	complete	average	average	corrosion	wood decay	leakage	corrosion
										condition	insect infestation	condition	condition	losses	insect infestation	decay	
										solid	reinforced brick (Klein)	brick	brick	wooden	purlin-column	asphalt roofing	galvanized sheet
80	23/90	Roosevelta 12	1905	#			5	C	S	good	weeping	satisfactory	satisfactory	poor	weeping	leakage	after renovation
										condition	cracks	condition	condition	condition	after renovation		
										solid	wooden	brick	brick	wooden	purlin-column	asphalt roofing	galvanized sheet
81	24/90	Kurkowa 36	1897	#			4	C	U	good	strengthened	average	average	weeping	wood decay	sealed	skewing
										condition	average condition	condition	condition	decay	wear of up to 30%		damaged

Key: Attic *: N—none, S—used as an attic, U—used as a flat. Basement **: N—none, CZ—partial, C—under the entire ground floor.

**Table 4 materials-14-02482-t004:** The probability of damage appearance related to 10 selected building elements.

The Probability of the Occurrence of Basic Damage p(di)			Foundations	Basement Walls	Solid Floors Above Basements	Structural Walls	Inter-Story Wooden Floors	Stairs	Roof structure	Window Joinery	Inner plasters	Facades
Group No	No. ofDamage	Name of Damage	p(d)2	p(d)3	p(d)4	p(d)7	p(d)8	p(d)9	p(d)10	p(d)13	p(d)15	p(d)20
A	d1	Mechanical damage						0.86		0.89	0.74	0.81
	d2	Leakage								0.93		
	d3	Brick losses	0.78	0.66		0.96		0.78				
	d4	Mortar losses		0.68	0.71	0.91						
	d5	Brick decay	0.76	0.63	0.73	0.79						
	d6	Mortar decay		0.49		0.74					0.80	0.86
	d7	Peeling off of paint coatings									0.79	0.85
	d8	Falling off of paint coatings									0.31	0.32
	d9	Cracks in bricks	0.78	0.52	0.56	0.25						
	d10	Cracks on plaster		0.39		0.69					0.78	0.82
	d11	Scratching on walls				0.11						
	d12	Scratching on plaster				0.55	0.80				0.73	0.75
	d13	Loosening of plaster					0.49				0.44	0.56
	d14	Falling off of plaster sheets									0.13	0.22
B	d15	Dampness	0.35	0.48	0.22	0.66	0.81		0.62	0.47	0.14	0.54
	d16	Weeping	0.09	0.24	0.18	0.18	0.53	0.59	0.44	0.29	0.1	0.38
	d17	Biological corrosion of brick	0.66	0.17		0.39						
	d18	Fungus					0.08				0.03	0.19
	d19	Rot and mold	0.03	0.05		0.05				0.11	0.03	0.14
	d20	Corrosion raid of steel beams			0.52			0.72				
	d21	Surface corrosion of steel beams			0.68			0.54				
	d22	Deep corrosion of steel beams			0.24			0.16				
	d23	Water flooding			0.04							
C	d24	Dynamic sensitivity of floor beams					0.64					
	d25	Deformations of wooden beams					0.35					
	d26	Skewing of window joinery								0.75		
	d27	Warping of window joinery								0.59		
	d28	Delamination of wooden elements							0.51			
D	d29	Partial insect infestation of wooden elements						0.07	0.15	0.10		
	d30	Complete insect infestation of wooden elements					0.25		0.5	0.30		

**Table 5 materials-14-02482-t005:** Point biserial correlation coefficients of overground structural walls for We, Wt*, Wt**.

		Point Biserial Correlation Coefficient for:								
W7	Structural Walls	Observed Wear (We) Corresponding to the II, III and IV Maintenance Conditions of an Element			Observed Wear (We), Theoretical: (Wt*) for T = T [8] and (Wt**) for T = t (max) in the I-V Maintenance Condition			Difference between the Average Value of the Observed Wear (We) and the Theoretical Wear (Wt*, Wt**) for Cases where u = 1 and u = 0 [%]		
No. of Damage	Name of Damage	r(We)II	r(We)III	r(We)IV	r(We)	r(Wt*)	r(Wt**)	DE(We)	DE(Wt*)	DE(Wt**)
d3	Brick losses	d(We) = 0	0.02	0.00	0.19	0.09	0.10	11.20	6.36	6.90
d4	Mortar losses	d(We) = 0	0.01	0.00	0.30	0.09	0.11	12.40	4.48	5.20
d5	Brick decay	d(We) = 0	0.05	0.45	0.17	0.08	0.09	4.90	2.65	2.97
d6	Mortar decay	d(We) = 0	0.06	0.10	0.09	−0.05	−0.04	2.30	−1.45	−1.20
d9	Cracks in bricks	d(We) = 0	0,00	0.03	0.11	0.10	0.09	2.80	3.18	3.00
d10	Cracks on plaster	d(We) = 0	0.07	0.29	0.03	0.03	0.03	0.69	0.80	0.80
d11	Scratching on walls	d(We) = 0	0.3	0.01	0.21	−0.02	−0.01	7.78	−0.77	−0.30
d12	Scratching on plaster	d(We) = 0	0.02	0.25	0.12	0.09	0.10	2.81	2.56	2.90
d15	Dampness	d(We) = 0	0.36	0.10	0.56	0.36	0.35	13.6	10.49	10.40
d16	Weeping	d(We) = 0	0.54	0.35	0.46	0.16	0.17	13.9	5.70	6.20
d17	Biological corrosion of bricks	d(We) = 0	0.62	0.01	0.67	0.28	0.31	16,0	8.17	8.80
d19	Mold and rot	d(We) = 0	0.18	0.31	0.34	0.09	0.09	18.1	5.82	5.90

Key: Observed wear Wt* for T = T [8] and Wt** for T = t (max) in the I-V maintenance condition.

**Table 6 materials-14-02482-t006:** Ranges of changes in the probabilities and correlations of damage and technical wear We, Wt*, Wt**.

W2-Foundations, W3-Basement walls, W4-Solid Floors above Basements, W7-Structural Walls, W8-Inter-Story Wooden Floors, W9-Stairs, W10-Roof Structure, W13-Window Joinery, W15-Inner Plasters, W20-Facades		The Probability of Damage to an Element, Which Corresponds to the II, III, IV Condition of the Element’s Maintenance						The Point Biserial Correlation Coefficient for the Observed Wear (We) and Theoretical Wear (Wt*), (Wt**)					
Number of Damage	Name of Damage	p(d) II		p(d) III		p(d) IV		r(We)		r(Wt*)		r(Wt**)	
		min	max	min	max	min	max	min	max	min	max	min	max
d7	Peeling off of paint coatings	0.59	0.90	0.76	0.91	0.86	0.96	0.15	**0.55**	0.20	0.40	0.02	0.43
d8	Falling off of paint coatings	0.00	0.20	0.23	0.31	0.23	0.30	0.25	**0.57**	0.07	0.25	0.13	0.30
d9	Cracks in bricks	0.06	0.65	0.30	0.83	0.28	0.55	0.01	0.11	0.00	0.24	0.00	0.15
d10	Cracks on plaster	0.25	0.75	0.44	0.86	0.29	1.00	0.03	**0.63**	0.03	0.44	0.03	**0.6**
d12	Scratching on plaster	0.30	0.75	0.39	0.92	0.75	1.00	0.05	**0.63**	0.09	0.47	0.10	**0.58**
d13	Loosening of plaster	0.00	0.44	0.14	0.57	0.63	0.95	0.09	**0.81**	0.09	0.33	0.25	**0.56**
d14	Falling off of plaster sheets	0.00	0.00	0.00	0.05	0.22	0.36	**0.50**	**0.57**	0.04	0.19	0.31	**0.50**
d15	Dampness	0.00	0.78	0.02	0.95	0.56	1.00	0.07	**0.84**	0.04	0.41	0.16	**0.59**
d16	Weeping	0.00	0.39	0.00	0.76	0.14	0.91	0.27	**0.79**	0.03	0.35	0.17	**0.59**
d17	Biological corrosion of bricks	0.00	0.00	0.00	0.31	0.25	0.84	0.31	**0.67**	0.22	0.28	0.2	0.23
d18	Fungus	0.00	0.00	0.00	0.08	0.00	0.63	0.38	**0.6**	0.02	0.18	0.23	0.48
d20	Corrosion raid of steel beams	0.06	0.13	0.60	0.82	0.7	0.91	0.42	**0.54**	0.13	0.31	0.19	0.22
d21	Surface corrosion of steel beams	0.06	0.52	0.45	0.71	0.78	0.88	0.29	**0.61**	0.00	0.40	0.07	0.40
d22	Deep corrosion of steel beams	0.00	0,00	0.06	0.22	0.28	0.43	**0.53**	**0.55**	0.04	0.18	0.26	0.29
d29	Partial insect infestation of wooden elements	0.00	0.08	0.02	0.10	0.09	0.19	0.28	0.45	0.00	0.20	0.18	**0.51**
d30	Complete insect infestation of wooden elements	0.00	0.07	0.33	0.43	0.63	0.77	0.42	**0.57**	0.00	0.30	0.38	**0.54**

Key: Observed wear Wt* for T = T [8] and Wt** for T = t (max) in the I-V maintenance condition.

**Table 7 materials-14-02482-t007:** The results of analyzing the relationship “damage—technical wear” using the correlation coefficient.

The Point Biserial Correlation Coefficient r(W) between the Measurable Variable (wi) and the Dichotomous Variable (di) in a Sample with a Size of 95 < n < 102		Foundations	Basement Walls	Solid Floors above Basements	Structural Walls	Inter-Story Wooden Floors	Stairs	Roof Structure	Window Joinery	Inner Plasters	Facades
No.of Damage	Name of Damage	r(W)2	r(W)3	r(W)4	r(W)7	r(W)8	r(W)9	r(W)10	r(W)13	r(W)15	r(W)20
d1	Mechanical damage						0.05		**0.29**	0.09	**0.28**
d2	Leaks								**0.26**		
d3	Brick losses	0.13	**0.23**	0.08	0.19		0.03				
d4	Mortar losses		**0.28**		**0.30**						
d5	Brick decay	0.14	0.07	0.00	0.17						
d6	Mortar decay		0.05		0.09					**0.47**	**0.48**
d7	Peeling off of paint coatings									0.15	**0.55**
d8	Falling off of paint coatings									**0.25**	**0.57**
d9	Cracks in bricks	0.05	0.01	0.05	0.11						
d10	Cracks on plaster		0.03		0.03					**0.3**	**0.63**
d11	Scratching on walls				**0.21**						
d12	Scratching on plaster				0.12	0.05				0.18	**0.63**
d13	Loosening of plaster					0.09				**0.67**	**0.81**
d14	Falling off of plaster sheets									**0.57**	**0.50**
d15	Dampness	**0.7**	**0.74**	**0.58**	**0.56**	0.07		**0.43**	**0.83**	**0.70**	**0.84**
d16	Weeping	**0.64**	**0.52**	**0.67**	**0.46**	**0.27**	**0.58**	**0.5**	**0.74**	**0.61**	**0.79**
d17	Biological corrosion of bricks	**0.36**	**0.31**		**0.67**						
d18	Fungus					**0.45**				**0.38**	**0.60**
d19	Mold and rot	**0.49**	**0.43**		**0.34**				**0.49**	**0.41**	**0.56**
d20	Corrosion raid of steel beams			**0.42**			**0.54**				
d21	Surface corrosion of steel beams			**0.29**			**0.61**				
d22	Deep corrosion of steel beams			**0.55**			**0.53**				
d23	Flooding with water			**0.45**							
d24	Dynamic sensitivity of floor beams					0.00					
d25	Deformations of wooden beams					0.12					
d26	Skewing of window joinery								**0.42**		
d27	Warping of window joinery								0.04		
d28	Delamination of wooden elements							0.07			
d29	Partial insect infestation of wooden elements						**0.38**	**0.28**	**0.45**		
d30	Complete insect infestation of wooden elements					**0.43**		**0.57**	**0.42**		
	the number of degrees of freedom:	100	93	93	100	100	100	100	100	97	100

Key: r(W)i—a lack of relationship between “d” and “w”. r(W)i—a tendency towards the relationship between “d” and “w” (0.05 < *p*(r) < 0.10). r(W)i—a strong relationship between “d” and “w” (*p*(r) > 0.10).

## Data Availability

No new data were created or analyzed in this study. Data sharing is not applicable to this article.

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
