# Peer review of "Correlation between Defects and Technical Wear of Materials Used in Traditional Construction"

_materials, 2021, doi:10.3390/ma14102482_

Round 1
Reviewer 1 Report
Dear authors,
This paper is focusing on a very interesting part of existing structures. This topic is especially important because European building stock is very old and it should be properly renovated. The crucial step for the proper renovation, rehabilitation and reconstruction is the condition assessment.
The idea of the paper is interesting and should be investigated more.
The structure of the paper is very well organized.
The Abstract part is too long. Some sentences should be deleted from the abstract part. I.e. the sentence ”This is one of the few cases in statistics when properties of various types are being correlated” is not needed.
The results are shown in a good way.
STAR documentation is good, but maybe it can be improved.
Conclusions are nicely presented.
I have several remarks.
Line 18 (typo) - there is too much space between the words the & occurred
51 – dot is missing at the end of the sentence
Figure 1 and a lot of other Figures are not mentioned in the text.
70 - …of 100,000 flats per year is expected. Where does this number come from?
Table 1 – where are the statistics for the buildings built after 1995?
Table 2 – why do you have 7 papers by Nowogońska? Please consider deleting some of them because they are dealing with the (more or less) same thing. You can include new research papers about the technical assessment of buildings by other authors, i.e.
Stepinac et al. https://www.mdpi.com/2076-3417/10/5/1576
Barbieri, G.; Biolzi, L.; Bocciarelli, M.; Fregonese, L.; Frigeri, A. Assessing the Seismic Vulnerability of a Historical Building. Eng. Struct.,
Borri, A.; Corradi, M.; De Maria, A.; Sisti, R. Calibration of a Visual Method for the Analysis of the Mechanical Properties of Historic Masonry. Procedia Struct. Integr. 2018
There is no obligation to have these research items added to your work but 7 papers from Nowogońska is also not needed. There is also 8 papers of self-citations. All of them are not needed. Please reduce the number of self-citations.
Table 3- it should be in an Annex. It is hard to read it here where it is.
256-269 I would rather see this in a graphical form. Please consider having a better interpretation of the data
297 A, B, C, D damages – Is there no single one damage because of unconsolidated settlement (subsidence)?
In conclusion, the paper is very well written but needs minor changes. My major concern is that it is very difficult to read because of a lot of tabulated data. It is often too demanding to read all of the numbers in the Tables. I would suggest having some infographics instead of tabulated data. Please try to organize at least some of the data in that way.
My 2nd major concern is that everything is based on a visual assessment subjected to a subjective interpretation of an examiner. I know that you are aware of that, but it should be pointed out. If the renovation of the building is undergoing, the full assessment should be done.
This paper is valuable for publishing it in the Journal.

Author Response
Wrocław, Poland, 27th April 2021
Dear Reviewer of Applied Sciences,
Thank you for the review of our paper materials-1189661entitled “Correlation Between Defects and Technical Wear of Materials Used in Traditional Construction” to be published in the journal Materials, Special Issue “Advanced Construction Materials and Processes in Poland”.
We appreciate your thoughtful and accurate comments as well as appreciation of our research works. We have carefully considered all comments and have now completed the revisions incorporating your suggestions in the revised uploaded manuscript.
We hope that the revised paper meets your expectations.
Kind regards,
Jarosław Konior and Mariusz Rejment
Department of Building Engineering, Faculty of Civil Engineering, Wroclaw University of Science and Technology, 50-370 Wrocław, Poland
Here are answers to reviewer’s comments:
REVIEWER 1
General Comments. Dear authors, This paper is focusing on a very interesting part of existing structures. This topic is especially important because European building stock is very old and it should be properly renovated. The crucial step for the proper renovation, rehabilitation and reconstruction is the condition assessment. The idea of the paper is interesting and should be investigated more. The structure of the paper is very well organized. STAR documentation is good, but maybe it can be improved. The results are shown in a good way. Conclusions are nicely presented. This paper is valuable for publishing it in the Journal.
Answer to General Comments. Thank you for your openness and appreciation of our research works. Indeed, the paper is devoted to the topical issue of ensuring the technical condition and maintenance of traditional residential buildings that have served for more than 100 years. In many countries around the world, maintaining the proper technical condition of the old housing stock is a problem that affects not only technical but also socio-cultural aspects. Many old houses reflect the cultural traditions of the country and have the status of a cultural heritage site. The article analyzes the influence of many factors causing their accelerated deterioration of buildings which are made of basic materials (cement, concrete, steel, timber, plaster, brick). The estimated durability is a major parameter of the changing age of the apartment buildings under study. Based on the probabilistic approach, methodological aspects of the technical assessment of the state of the structure have been developed, which should also be aimed at minimizing the subjectivity of expert assessment in the process of technical inspections of residential buildings.
Comment 1. The Abstract part is too long. Some sentences should be deleted from the abstract part. I.e. the sentence ”This is one of the few cases in statistics when properties of various types are being correlated” is not needed.
Answer 1. Good point, thank you. The abstract is too detailed and has been simplified and shortened accordingly.
Comment 2. Line 18 (typo) - there is too much space between the words the & occurred.
Answer 2. Indeed, one space between these words was unnecessary. Corrected.
Comment 3. 51 – dot is missing at the end of the sentence.
Answer 3. Indeed, dot was missing. Added.
Comment 4. Figure 1 and a lot of other Figures are not mentioned in the text.
Answer 4. There are only two figure in the paper. Corrected and now both figures have appropriate reference in the text.
Comment 5. 70 - …of 100,000 flats per year is expected. Where does this number come from?
Answer 5. This is only supplementary information which does not affect the methods and findings of the paper but provides the reader the scale of historical data. A gradual increase in residential investments to the level of 100,000 flats per year is expected. The number comes from the last "Population and Housing Census in Wroclaw", which was carried out using the representative method in 2015, and which was prepared and published in 2016 [1] - Tab. 1.
Comment 6. Table 1 – where are the statistics for the buildings built after 1995?
Answer 6. Sorry, it was mistake in heading of columns. The last one reflects the period of 1996 – 2010. Corrected in the Table 1.
Comment 7. Table 2 – why do you have 7 papers by Nowogońska? Please consider deleting some of them because they are dealing with the (more or less) same thing. You can include new research papers about the technical assessment of buildings by other authors, i.e.
- Stepinac et al. https://www.mdpi.com/2076-3417/10/5/1576
- Barbieri, G.; Biolzi, L.; Bocciarelli, M.; Fregonese, L.; Frigeri, A. Assessing the Seismic Vulnerability of a Historical Building. Eng. Struct.,
- Borri, A.; Corradi, M.; De Maria, A.; Sisti, R. Calibration of a Visual Method for the Analysis of the Mechanical Properties of Historic Masonry. Procedia Struct. Integr. 2018
There is no obligation to have these research items added to your work but 7 papers from Nowogońska is also not needed. There is also 8 papers of self-citations. All of them are not needed. Please reduce the number of self-citations.
Answer 7. Well, Nowogońska seems to be a “guru” on technical assessment of existing buildings and her works are valuable to my research. However, point taken and Nowogońska references have been reduced to the most essential 2 items. As far as the main author is concerned, the self-citations have not been incorporated due to any publishing benefits but to prove the logical and a long term continuity of my and co-authors research works on the subject. We are also referring to the previous findings in the paper. However, point taken and Konior references have been reduced to 1 introducing item and 1 item just published in Materials: Konior, J. “Overdurability and technical wear of materials used in the construction of old buildings”. Materials. 2021, vol. 14, 2, pp. 1-21, https://doi.org/ 10.3390/ma14020378. We have carefully analysed 3 suggested publications and do not feel that they are relevant to the presented topic of Correlation Between Defects and Technical Wear. Two first of them are specifically related to seismic conditions and masonry structures and the third one is about calibration of method for mechanical properties analysis, also related to masonry structure. If we incorporated them in our manuscript body it would make it more inconsistent and not very much comprehensive, so we’d better not if you do not mind.
Comment 8. Table 3- it should be in an Annex. It is hard to read it here where it is.
Answer 8. Yes, this is true. Table 3 is hardly readable, so we have reformatted it in one merged table of descriptive analysis presented in one page landscape format.
Comment 9. 256-269 I would rather see this in a graphical form. Please consider having a better interpretation of the data
Answer 9. Good point. Basic statistical values of the technical wear of 23 building elements have been presented in more visible, graphical form of figure 3.
Comment 10. 297 A, B, C, D damages – Is there no single one damage because of unconsolidated settlement (subsidence)?
Answer 10. Yes, there are 30 single damages. The result of an in-depth analysis of the examined group of 102 downtown tenement houses involved the identification of all the individual damage to their building elements at the elementary level. The synthesis of the type of elementary damage with regards to the common source of their formation (B, D), and the similarity of the consequences caused by them (A, B, C), enabled the damage to be semantically and generically ordered into the following groups: A, B, C, D. The detailed analysis of the characteristics of the damage to elements of downtown tenement houses takes into account its four-level classification at the level of general synthesis. Moreover, the analysis separates generically integrated elementary damage, and due to this the number of types of damage increases to 30. It also includes a very important assumption, according to which the fact of the occurrence of the identified damage is described by number 1, and its absence by 0. In the detailed characteristics of damage, it was possible to assign each elementary damage to its element, which was described by the degree of its technical wear. Therefore, the first step of the transition from a qualitative to quantitative notation in the construction of the cause-effect model "damage - technical wear" was made.
Comment 11. In conclusion, the paper is very well written but needs minor changes. My major concern is that it is very difficult to read because of a lot of tabulated data. It is often too demanding to read all of the numbers in the Tables. I would suggest having some infographics instead of tabulated data. Please try to organize at least some of the data in that way.
Answer 11. Yes, we are aware that many tabulated date may be difficult to digest for the reader, therefore we converted some tables into infographics (e.g. tab 4 into fig 3 with a clear graphical form); also other tables have been simplified and better formatted.
Comment 12. My 2nd major concern is that everything is based on a visual assessment subjected to a subjective interpretation of an examiner. I know that you are aware of that, but it should be pointed out. If the renovation of the building is undergoing, the full assessment should be done.
Answer 12. Yes, absolutely agreed. The entire methodology (probabilistic and fuzzy) of the detailed assessment (theoretical and observed) of residential buildings to be renovated has been presented in our previous publications, as follows:
- Konior, J. Decision assumptions on building maintenance management. Probabilistic methods, Arch. Civ. Eng., 2007, 53, pp. 403–423.
- Konior, J. Technical assessment of old buildings by fuzzy approach, Archives of Civil Engineering, 2019, 65(1), pp.129–142, https://doi.org/10.2478/ace-2019-0009.
- Konior, J. Technical Assessment of old buildings by probabilistic approach., Archives of Civil Engineering, 2020, 66(3), pp. 443–466, https://doi.org/10.24425/ace.2020.134407.
- Konior, J. Maintenance of apartment buildings and their measurable deterioration, Tech. Trans. Czas. Tech. 2017, 6, pp. 101–107, https://doi.org/10.4467/2353737xct.17.090.6566.
- Konior, J. Bi-serial correlation of civil engineering building elements under constant technical deterioration, J. Sci. Gen. Tadeusz Kosciuszko Mil. Acad. L. Forces. 2016, 179, pp. 142–150.
- Konior, J. Intensity of defects in residential buildings and their technical wear, Tech. Trans. Civ. Eng. 2014, 111(2-B), pp. 137–146.
- Konior, J.; Sawicki, M.; Szóstak, M. Influence of Age on the Technical Wear of Tenement Houses. Appl. Sci. 2021, 11, 297. https://doi.org/10.3390/app11010297
- Konior, J.; Sawicki, M.; Szóstak, M. Intensity of the Formation of Defects in Residential Buildings with Regards to Changes in Their Reliability. Appl. Sci. 2020, 10, 6651. https://doi.org/10.3390/app10196651
- Konior, J.; Sawicki, M.; Szóstak, M. Damage and Technical Wear of Tenement Houses in Fuzzy Set Categories. Appl. Sci. 2021, 11, 1484. https://doi.org/10.3390/app11041484
- Konior, J. Overdurability and technical wear of materials used in the construction of old buildings. Materials. 2021, vol. 14, 2, pp. 1-21, https://doi.org/ 10.3390/ma14020378
- Konior, J.; Stachoń, T. Fuzzy relations matrixes of damages and technical wear related to apartment houses. Applied Sciences. 2021, vol. 11, 5, pp. 1-14., https://doi.org/10.3390/app11052223
- Konior, J.; Stachoń, T. Bayes conditional probability of fuzzy damage and technical wear of residential buildings. Applied Sciences. 2021, vol. 11, 6, pp. 1-23.https://doi.org/ 10.3390/app11062518
Many models and tools have been presented and validated as to lessen the subjectivity of an examiner / expert. While assessing building elements’ technical wear - apart from applying the measurable (qualitative) criteria - the immeasurable (quantitative) criteria representing symptoms (pinpointed defects) of their deterioration have been taken into account. Only very few of these criteria can be classified at high level of probability. There are symptoms of extreme characters, described by extreme dichotomic divisions. It is, however, agreed that between e.g. a total pest attack to wooden elements and its lack the mid-states appear. Their value is often appreciate in verbal way, e.g. “substantially”, considerably”, “significantly”, “partially”, “hardly” and it is always met in a description of detected defects as a result of a building objects technical inspections. Therefore, the research led towards looking at the problem from the angle, which gave right to describe naturally qualitative variables (so immeasurable) and determine existing conditional probabilities in fuzzy sets categories. This is novel approach and as fuzziness supplements the random methodology developed by author’s in previous papers. Here, to fulfil the purpose of the study, we concentrate on correlation between defects and technical wear of materials used in traditional construction and its transfer to a general population that constitutes around 20% of housing in Wrocław which is a challenging issue.

Reviewer 2 Report
This reviewer is having difficulty expressing an opinion on this paper. In fact, although the paper contains a lot of interesting information that allowed the reviewer to broaden the knowledge of Polish construction techniques in the second half of the 19th century, up to the outbreak of the First World War, the content of the paper does not seem to be fully relevant to the Aims & Scope of “Materials”. The reviewer suggests submitting the paper to some more relevant MDPI journal, such as “Construction Materials”.
Author Response
Wrocław, Poland, 27th April 2021
Dear Reviewer of Applied Sciences,
Thank you for the review of our paper materials-1189661entitled “Correlation Between Defects and Technical Wear of Materials Used in Traditional Construction” to be published in the journal Materials, Special Issue “Advanced Construction Materials and Processes in Poland”.
We appreciate your thoughtful and accurate comments as well as appreciation of our research works. We have carefully considered all comments and have now completed the revisions incorporating your suggestions in the revised uploaded manuscript.
We hope that the revised paper meets your expectations.
Kind regards,
Jarosław Konior and Mariusz Rejment
Department of Building Engineering, Faculty of Civil Engineering, Wroclaw University of Science and Technology, 50-370 Wrocław, Poland
Here are answers to reviewer’s comments:
REVIEWER 2
General Comments. This reviewer is having difficulty expressing an opinion on this paper. In fact, although the paper contains a lot of interesting information that allowed the reviewer to broaden the knowledge of Polish construction techniques in the second half of the 19th century, up to the outbreak of the First World War, the content of the paper does not seem to be fully relevant to the Aims & Scope of “Materials”. The reviewer suggests submitting the paper to some more relevant MDPI journal, such as “Construction Materials”.
Answer to General Comments. We do appreciate the reviewer’s valuable view on the presented topic but cannot agree to full extant. Firstly, because the paper and the subject represents a special contribution to journal of Materials requested by the editor to submit. Secondly, because topic of the research concerns old residential buildings that consist of basic materials (cement, concrete, steel, timber, plaster, brick) using traditional technology and reflects the aim and scope of Materials. The practical results for a representative group of old downtown apartment houses with traditional structure, erected in Wrocław (Poland) at the turn of the 19th and 20th centuries, can be therefore summed up by the following conclusions:
- Age of elements of old apartment houses with a traditional structure:
- is of secondary importance in the process of the intensity of loss of its useful values;
- is not the essential size determining the course of their technical wear and tear;
- The technical extent of wear and tear of the components of an old residential building is determined by the conditions for its maintenance and use;
- The previous theoretical methods for measuring the technical wear and tear of the building and its components do not sufficiently describe the actual states, which is called into question:
- how these methods are assigned to the maintenance conditions of the building;
- not precise selection of too general forms of mathematical functions;
- Quantitative damage analysis carried out by experiential methods of assessing the technical condition of the building shall indicate the nature and magnitude of the damage to its components which are characteristic of the relevant maintenance conditions;
- A study analysis of the processes of operation of residential objects and the basic dependencies of reliability theory made in it indicates that for the useful life of an object in which the working time to damage has an exponential distribution (this is in principle the life expectancy corresponding to the length of service of the dwellings concerned), the average remaining time of unsafe operation is constant at all times. Theoretically, therefore, residential objects, after some time of trouble-free operation, perform their functions as new. The age of the elements of an old residential building is then of secondary importance in the process of the intensity of loss of its useful value;
- If assumed that the measure of matching the nonlinear mathematical models tested in the nonlinear regression method, as a function of the technical consumption of building elements over time, is the determination factor, then no more than 30% of the destruction of the elements is explained by the passage of time. Age is therefore not a determinant of the technological consumption of the elements of the buildings analysed;
- Previous theoretical methods for measuring the technical wear and tear of the building and its components do not reflect the actual course of the deterioration process over time. Two facts pay attention:
- an assessment of the significance of the differences between the theoretical and observed technical consumption distribution values of building elements by WILCOXON test and the Sign Test in most cases confirmed the conclusions of their comparative analysis and showed the significance of the differences between the distributions of theoretical and observed wear, although the Sign Test indicated their identity in the case of foundation distributions, underground walls and structural walls in the all five conditions of maintenance, while both WILCOXON test and the Sign Test confirmed the identity of the distributions only in individual maintenance groups of the building;
- adopting too general and not always appropriate forms of parabolic and linear functions to describe the theoretical side of the progress of the technical consumption of the building's elements with age; of the four nonlinear regression separable tested, new mathematical models, none of the power (parabolic) models represent the nature of the designated trend of the time-consuming process (very low determination factor and unnatural size of parameterized durability); analysis of variance in the nonlinear regression method also indicates a much better representation of the modelled trend by exposive and hyperbolic dependencies and slightly worse by linear functions;
- The quantitative analysis of damage, carried out by empirical (visual) methods of assessing the technical condition of the buildings, indicates the type and determines the magnitude of these damage to its components which are characteristic of the appropriate conditions of maintenance. Studies of cause - effect "damage - technical wear" in observed states allow a numerical recognition of the impact of the building's maintenance conditions on the degree of technical wear of its components:
- the direction of the relationship is right-hand (positive) for all test elements of the building, but the correlation force between the defect occurring and their technical wear shows a significant span (from 0.00 to 0.84) depending on the conditions of the buildings maintenance;
- the rule is that correlations of at least moderate strength always show damage caused by water penetration and moisture penetration (on average 0,54); only in the case of internal plasters and façade, individual mechanical damage to their structure and texture can also be considered moderate and quite strong;
- for the accepted confidence level of 95%, the dependence of moderate force can be applied to 34-48% of the general population size, and the correlations quite strong - to 49-71%.
Therefore, we reckon that our research works are coherent, consistent, comprehensive, solid and reliable and may be of Materials interest.

Reviewer 3 Report
The paper presents the correlation between defects and technical wear of materials used in traditional construction in Wroclaw, Poland. My major critiques go to lack of conclusion. The paper should have a conclusion section. The paper also lacks proper visualization for findings in a clearer manner. It is a bit difficult to pick the key findings easily from some of the Tables. The authors could represent them in the form of figures (heatmaps). Some tables are not easy to understand. For instance, in Table 3, there are two rows in each case (from column number 11 to 18) and it is not clear how to read them. The authors are highly encouraged to take the following minor comments into consideration and revise the manuscript accordingly.
- The paper is based on the case studies and this fact shall be reflected in the title.
- The abstract is a bit lengthy and needs to be shortened.
- Replace commas by full stop in the value of the correlation of coefficients, e.g. line 24, 26
- Be consistent in refereeing Tables and Figures in the body part of the article, e.g. in line 96 Tab. 1 should read as Table 1.
- There are several parts in which words are not fully visible in the Tables. E.g. header of column four in Tables 3 which says “year of”. There is something missed. Check all phrases in all tables and make all visible.
- There is an error in the key presented under Table 8.
- Avoid unnecessary indentation for subsection 2.5.3
- The page numbers after each section should not start as a new page. It must be connected from the above section in order to read correct.
Author Response
Wrocław, Poland, 27th April 2021
Dear Reviewer of Applied Sciences,
Thank you for the review of our paper materials-1189661entitled “Correlation Between Defects and Technical Wear of Materials Used in Traditional Construction” to be published in the journal Materials, Special Issue “Advanced Construction Materials and Processes in Poland”.
We appreciate your thoughtful and accurate comments as well as appreciation of our research works. We have carefully considered all comments and have now completed the revisions incorporating your suggestions in the revised uploaded manuscript.
We hope that the revised paper meets your expectations.
Kind regards,
Jarosław Konior and Mariusz Rejment
Department of Building Engineering, Faculty of Civil Engineering, Wroclaw University of Science and Technology, 50-370 Wrocław, Poland
Here are answers to reviewer’s comments:
REVIEWER 3
General Comments. The paper presents the correlation between defects and technical wear of materials used in traditional construction in Wroclaw, Poland. My major critiques go to lack of conclusion. The paper should have a conclusion section. The paper also lacks proper visualization for findings in a clearer manner. It is a bit difficult to pick the key findings easily from some of the Tables. The authors could represent them in the form of figures (heatmaps). Some tables are not easy to understand. For instance, in Table 3, there are two rows in each case (from column number 11 to 18) and it is not clear how to read them. The authors are highly encouraged to take the following minor comments into consideration and revise the manuscript accordingly.
Answer to General Comments. The critique is constructive and the minor comments have already been addressed the following way:
- table 3 has been reformatted and now one address presented in a row is equivalent to one technical description of building elements presented in columns 11 – 8
- table 4 presenting the statistical data has been converted into the figure 3 with a clear picture and a vivid comparison of stats calculations in a graph form
- tables 4 – 8 have been reformatted
- table 9 has been deleted and its main findings have been presented in the relevant item 2.5.3
- new conclusions have been added to the Discussion, section 4:
“Reliability and novelty of the research has been proved by new approach to unique correlations of two different kinds random and was published in the papers applying probabilistic and correlation methodology. The correctness of the test results for a representative group of old downtown apartment houses with traditional structure, erected in Wrocław (Poland) at the turn of the 19th and 20th centuries, can be therefore summed up by the following conclusions:
- age of elements of old apartment houses with a traditional structure:
- is of secondary importance in the process of the intensity of loss of its useful values;
- is not the essential size determining the course of their technical wear and tear;
- the technical extent of wear and tear of the components of an old residential building is determined by the conditions for its maintenance and use;
- quantitative damage analysis carried out by experiential methods of assessing the technical condition of the building shall indicate the nature and magnitude of the damage to its components which are characteristic of the relevant maintenance conditions;
- a study analysis of the processes of operation of residential objects and the basic dependencies of reliability theory made in it indicates that for the useful life of an object in which the working time to damage has an exponential distribution (this is in principle the life expectancy corresponding to the length of service of the dwellings concerned), the average remaining time of unsafe operation is constant at all times. Theoretically, therefore, residential objects, after some time of trouble-free operation, perform their functions as new. The age of the elements of an old residential building is then of secondary importance in the process of the intensity of loss of its useful value;
- if assumed that the measure of matching the nonlinear mathematical models tested in the nonlinear regression method, as a function of the technical consumption of building elements over time, is the determination factor, then no more than 30% of the destruction of the elements is explained by the passage of time; age is therefore not a determinant of the technological consumption of the elements of the buildings analysed.”
Comment 1. The paper is based on the case studies and this fact shall be reflected in the title.
Answer 1. Indeed, the paper is based on the case studies but we do not share the opinion this fact shall be reflected in the title, we are afraid. The subject of the research involves tenement houses in a separate part of the downtown district in Wroclaw, Poland. The buildings are situated along downtown streets of secondary importance in an urban layout that has remained unchanged for years. They are front buildings and also outbuildings with a modest architectural design and economical functional standard. The facilities were built of brick in longitudinal, usually three‐bay, structural systems. 102 tenement houses were mainly erected in the second half of the nineteenth century, until the outbreak of World War I. The above-described downtown residential buildings with construction and material solutions typical for the turn of the 19th and 20th centuries, similar functions and standards, and a specific form of ownership (the so-called pre-war "tenement houses") are defined by the term "tenement houses". The research sample, covering 102 technically assessed residential buildings from Wroclaw's Srodmiescie district, was selected from a group of 160 examined buildings. The overriding criterion for the selection of the sample was the obtaining of a comparable group of objects. Mutual comparability of downtown tenement houses meant:
- age coherence, i.e. a similar period of erection, maintenance and exploitation with regards to historical and social aspects;
- compact development in the urban layout that has remained unchanged for years;
- similar location along downtown street routes with an urban, but not representative, character;
- construction and material homogeneity, especially regarding the load-bearing structure of buildings;
- identical functional solutions, which are understood as the standard of apartment amenities and furnishings in force at that time, and also a specific standard of living of residents.
A method of selecting the research sample at the level of greater detail was based on the mutual similarity of all technical solutions of downtown tenement houses. The selected research sample, according to the criteria presented above, is representative with regards to one of the concepts (specific for the adopted purpose of the study) of representativeness. It contains all the values of the variables, which could be recreated from previous research that had a different objective function than the one adopted in this study. However, these values were compiled and processed in such a way that it is possible to make conclusions about the cause-effect relationships between them in the general population. Thus, the typological representativeness of the sample into which the desired types of homogeneous variables are classified can be assumed. Due to the fact that the structure of the population and its properties were well recognized earlier, such a selection of the research sample can also be considered to be deliberate. It should be noted that the sample may not be representative in terms of the distributions of the examined variables, which may - for the adopted significance level - not correspond to the analogous distributions in the general population. It is also not known - at this stage of the research - whether the selected sample is representative due to the correspondence between its variables and the identically defined variables in the entire set of downtown residential buildings.
Therefore, the investigated residential houses stand for significant part of tenement buildings which in Wroclaw city constitutes like 20% of the overall houses and the general title of the paper is sufficiently justified.
Comment 2. The abstract is a bit lengthy and needs to be shortened.
Answer 2. Good point, thank you. The abstract is too detailed and has been simplified and shortened accordingly.
Comment 3. Replace commas by full stop in the value of the correlation of coefficients, e.g. line 24, 26
Answer 3. Absolutely right but these numbers appear in the part of the abstract which has been deleted.
Comment 4. Be consistent in refereeing Tables and Figures in the body part of the article, e.g. in line 96 Tab. 1 should read as Table 1.
Answer 4. All Tables and Figures have been referred in a full wording, not abbreviations.
Comment 5. There are several parts in which words are not fully visible in the Tables. E.g. header of column four in Tables 3 which says “year of”. There is something missed. Check all phrases in all tables and make all visible.
Answer 5. All wording, phrases in table have been corrected as to make them visible.
Comment 6. There is an error in the key presented under Table 8.
Answer 6. Indeed, thanks for spotting. Corrected as follows: r(W)i – a strong relationship between "d" and "w" (p (r) > 0.10)
Comment 7. Avoid unnecessary indentation for subsection 2.5.3
Answer 7. Fine but we do not see unnecessary indentation in the subsection 2.5.3
Comment 8. The page numbers after each section should not start as a new page. It must be connected from the above section in order to read correct.
Answer 8. Absolutely right. This was due to cutting a new page for tables. Corrected and now the body of the text has sequential numbering of pages.

Reviewer 4 Report
Ms. Ref. No.: Materials_ Manuscript ID_ materials-1189661 – peer-review
Correlation Between Defects and Technical Wear of Materials Used in Traditional Construction
Reviewer comments:
SUMMARY
The manuscript deals with an investigation on the correlation between defects and technical wear of materials used in traditional construction. This is a topic that has not been widely covered in the literature, therefore, this a subject of great interest, but it is somehow limited in the analysis and application of these results.
MAIN IMPRESSIONS
This paper has an undeniable practical usefulness. However, from a scientific point of view, the following issues must be addressed: i) Research results should be discussed in deep, i. e., it is necessary to support your findings with the previous literature ii) you should discuss your findings with other studies, iii) Conclusion is not mandatory when the discussion is long or complex, However, your discussion is weak, iv) the novelty of the paper should be underlined.
MORE DETAILED COMMENTS
Lines 8-30: The Abstract should be a single paragraph of about 200 words maximum.
Page numbering o wrong from page 6.
Format is wrong in several pages such as 6, 8 and son.
Line 65: I suggest to comment the EU's energy and environmental goals. To boost energy performance of buildings, the EU has established a legislative framework that includes the Energy Performance of Buildings Directive 2010/31/EU (EPBD) and the Energy Efficiency Directive 2012/27/EU (DIRECTIVE (EU) 2018/844 OF THE EUROPEAN PARLIAMENT AND OF THE COUNCILof 30 May 2018amending Directive 2010/31/EU on the energy performance of buildings and Directive 2012/27/EU on energy efficiency).
Lines 154: Table 2: I suggest to use these references to disscuss the paper.
Lines 155-170: I suggest to use these references to disscuss the paper.
Lines 578-618: Research results should be discussed in deep.
Lines 535 & 578: The paper has a page of results and a page of disscussion of 34. Then, it seems to be unbalanced.
Lines 618: Conclusions are missing.
RECOMMENDATION
In conclusion, Major changes have been proposed.
Author Response
Wrocław, Poland, 27th April 2021
Dear Reviewer of Applied Sciences,
Thank you for the review of our paper materials-1189661entitled “Correlation Between Defects and Technical Wear of Materials Used in Traditional Construction” to be published in the journal Materials, Special Issue “Advanced Construction Materials and Processes in Poland”.
We appreciate your thoughtful and accurate comments as well as appreciation of our research works. We have carefully considered all comments and have now completed the revisions incorporating your suggestions in the revised uploaded manuscript.
We hope that the revised paper meets your expectations.
Kind regards,
Jarosław Konior and Mariusz Rejment
Department of Building Engineering, Faculty of Civil Engineering, Wroclaw University of Science and Technology, 50-370 Wrocław, Poland
Here are answers to reviewer’s comments:
REVIEWER 4
General Comments. The manuscript deals with an investigation on the correlation between defects and technical wear of materials used in traditional construction. This is a topic that has not been widely covered in the literature, therefore, this a subject of great interest, but it is somehow limited in the analysis and application of these results. In conclusion, major changes have been proposed.
Answer to General Comments. Thank you for appreciation of value of or research works. Please note that the presented topic of biserial correlations between damages and technical wear is a continuation of the set of publications on the subject both by probabilistic and fuzzy approaches. The further analysis and applications have been presented in the following papers which we do not refer here as to avoid self-citations:
- Konior, J. Decision assumptions on building maintenance management. Probabilistic methods, Arch. Civ. Eng., 2007, 53, pp. 403–423.
- Konior, J. Technical assessment of old buildings by fuzzy approach, Archives of Civil Engineering, 2019, 65(1), pp.129–142, https://doi.org/10.2478/ace-2019-0009.
- Konior, J. Technical Assessment of old buildings by probabilistic approach., Archives of Civil Engineering, 2020, 66(3), pp. 443–466, https://doi.org/10.24425/ace.2020.134407.
- Konior, J. Maintenance of apartment buildings and their measurable deterioration, Tech. Trans. Czas. Tech. 2017, 6, pp. 101–107, https://doi.org/10.4467/2353737xct.17.090.6566.
- Konior, J. Bi-serial correlation of civil engineering building elements under constant technical deterioration, J. Sci. Gen. Tadeusz Kosciuszko Mil. Acad. L. Forces. 2016, 179, pp. 142–150.
- Konior, J. Intensity of defects in residential buildings and their technical wear, Tech. Trans. Civ. Eng. 2014, 111(2-B), pp. 137–146.
- Konior, J.; Sawicki, M.; Szóstak, M. Influence of Age on the Technical Wear of Tenement Houses. Appl. Sci. 2021, 11, 297. https://doi.org/10.3390/app11010297
- Konior, J.; Sawicki, M.; Szóstak, M. Intensity of the Formation of Defects in Residential Buildings with Regards to Changes in Their Reliability. Appl. Sci. 2020, 10, 6651. https://doi.org/10.3390/app10196651
- Konior, J.; Sawicki, M.; Szóstak, M. Damage and Technical Wear of Tenement Houses in Fuzzy Set Categories. Appl. Sci. 2021, 11, 1484. https://doi.org/10.3390/app11041484
- Konior, J. Overdurability and technical wear of materials used in the construction of old buildings. Materials. 2021, vol. 14, 2, pp. 1-21, https://doi.org/ 10.3390/ma14020378
- Konior, J.; Stachoń, T. Fuzzy relations matrixes of damages and technical wear related to apartment houses. Applied Sciences. 2021, vol. 11, 5, pp. 1-14., https://doi.org/10.3390/app11052223
- Konior, J.; Stachoń, T. Bayes conditional probability of fuzzy damage and technical wear of residential buildings. Applied Sciences. 2021, vol. 11, 6, pp. 1-23.https://doi.org/ 10.3390/app11062518
Comment 1. This paper has an undeniable practical usefulness. However, from a scientific point of view, the following issues must be addressed:
- research results should be discussed in deep, i. e., it is necessary to support your findings with the previous literature
- you should discuss your findings with other studies,
- conclusion is not mandatory when the discussion is long or complex, However, your discussion is weak,
- the novelty of the paper should be underlined.
Answer 1.
Re supporting findings with the previous literature - the following discussions of other findings are proposed: “The main scope of the work was the conducting of a qualitative analysis of damage to the elements of the inspected tenement houses. The technical characteristics and typological ordering of this damage, understood as an expression of the quality of maintenance of residential buildings, made possible to identify association of appearing defects and technical wear (if such a relationship exists). Therefore, an attempt was made to numerically express the relationship between the occurred defects of buildings and the extent of their technical wear process. When calculating the strength of this relationship, the method of determining the point biserial correlation coefficient for the measurable property and the dichotomous property was used. This is one of the very few cases in statistics when properties of various types are being correlated. A number of works by Nowogońska [16-20] were studied in the methodical approach to the technical assessment of tenement houses but none of them concentrated on cause (damage) – effect (maintenance) model. The fuzzy approach presented in the publications of Plebankiewicz, Wieczorek, and Zima [21-26] was used in the assessment of the whole service life of a building objects but considered the cost aspects in their life cycle. The social, cultural and engineering diagnosis of the purposefulness of the revitalization or demolition of buildings is presented in an interesting way in papers by Noonan, Power, Ástmarsson, at al. [2-6]. All of them are mainly descriptive and do not present the results of damage measurements. The life cycle of a building structure is presented within the engineering approach in articles of Chen, Frangopol, Silva, Grant, Saleh [23-27] but all of them are representing similar approach to Plebankiewicz with regards to cost approach in papers [28-33]. Research methods and models with a full methodology of cause-effect relationships ("damage - technical wear") were presented in publications of Kapliński and Konior [33-39], where methods and models for associating damage and technical wear were presented and discussed. Valuable works of Lee and Terentyev have also been analysed and taken into account as supplement with an approach to risk identification in the diagnosis of buildings [40-41].”
Re conclusions improvements - new conclusions have been added to the Discussion, section 4: “Reliability and novelty of the research has been proved by new approach to unique correlations of two different kinds random and was published in the papers applying probabilistic and correlation methodology. The correctness of the test results for a representative group of old downtown apartment houses with traditional structure, erected in Wrocław (Poland) at the turn of the 19th and 20th centuries, can be therefore summed up by the following conclusions:
- age of elements of old apartment houses with a traditional structure:
- is of secondary importance in the process of the intensity of loss of its useful values;
- is not the essential size determining the course of their technical wear and tear;
- the technical extent of wear and tear of the components of an old residential building is determined by the conditions for its maintenance and use;
- quantitative damage analysis carried out by experiential methods of assessing the technical condition of the building shall indicate the nature and magnitude of the damage to its components which are characteristic of the relevant maintenance conditions;
- a study analysis of the processes of operation of residential objects and the basic dependencies of reliability theory made in it indicates that for the useful life of an object in which the working time to damage has an exponential distribution (this is in principle the life expectancy corresponding to the length of service of the dwellings concerned), the average remaining time of unsafe operation is constant at all times. Theoretically, therefore, residential objects, after some time of trouble-free operation, perform their functions as new. The age of the elements of an old residential building is then of secondary importance in the process of the intensity of loss of its useful value;
- if assumed that the measure of matching the nonlinear mathematical models tested in the nonlinear regression method, as a function of the technical consumption of building elements over time, is the determination factor, then no more than 30% of the destruction of the elements is explained by the passage of time; age is therefore not a determinant of the technological consumption of the elements of the buildings analysed.”
Comment 2. Lines 8-30: The Abstract should be a single paragraph of about 200 words maximum.
Answer 2. Good point, thank you. The abstract is too detailed and has been simplified and shortened accordingly.
Comment 2. Page numbering o wrong from page 6. Format is wrong in several pages such as 6, 8 and son.
Answer 2. Absolutely right. This was due to cutting a new page for tables. Corrected and now the body of the text has sequential numbering of pages.
Comment 3. Line 65: I suggest to comment the EU's energy and environmental goals. To boost energy performance of buildings, the EU has established a legislative framework that includes the Energy Performance of Buildings Directive 2010/31/EU (EPBD) and the Energy Efficiency Directive 2012/27/EU (DIRECTIVE (EU) 2018/844 OF THE EUROPEAN PARLIAMENT AND OF THE COUNCIL of 30 May 2018 amending Directive 2010/31/EU on the energy performance of buildings and Directive 2012/27/EU on energy efficiency).
Answer 3. Thank you for paying us attention to the new EU’s legislation that will definitely affect our research works in terms of energy performance of buildings to be renovated. Therefore, while expressing directions for the future assessment of residential houses, the following text has been inserted in the section 4 Discussion and Conclusions:
“The authors of the article point out that that the new EU Energy Efficiency Directive EU/2018/844 of 30 May 2018 (amending Directive 2010/31/EU on the energy performance of buildings and Directive 2012/27/EU on energy efficiency) introduces a number of changes, new requirements and simplifications to some of the current rules mainly on energy consumption and sets out requirements for the renovation of existing buildings. According to the Directive, EU Members have to develop long-term strategies for the renovation of existing residential and non-residential, public and private buildings. The aim of this strategy is to target multi-annual actions on the renovation of existing buildings. The renovation strategy will require the introduction of new legal and technical regulations as well as new systems to support the implementation of renovations. According to the authors of the article, the data and results of their research work can help the Polish government to introduce new technical requirements for the renovation of existing multi-family residential buildings in Poland.”
Comment 3. Lines 154: Table 2: I suggest to use these references to discuss the paper.
Answer 4. Good suggestion. The references presented in the Table 2 are proposed to be used for discussion the paper – see answer 1.
Comment 4. Lines 155-170: I suggest to use these references to discuss the paper.
Answer 4. Good suggestion. The references presented in the Table 2 are proposed to be used for discussion the paper – see answer 1.
Comment 5. Lines 578-618: Research results should be discussed in deep. Lines 535 & 578: The paper has a page of results and a page of discussion of 34. Then, it seems to be unbalanced. Lines 618: Conclusions are missing.
Answers, 5. Research results have been presented and interpreted in conjunction with measurable values in the section 3: “The results of the investigated cause-and-effect relationship ("damage - technical wear") in the representative sample of downtown residential buildings erected using traditional methods allow for the formulation of the following conclusions - Tables 6 - 8:
- as a rule, damage caused by water penetration and moisture penetration (group II) has the greatest impact on the amount of technical wear of the elements;
- the correlation of at least moderate strength is always shown by damage caused by water penetration and moisture penetration (group II); only in the case of internal plasters and facades, can single mechanical damage to their structure and texture be considered moderate and quite strong (group I);
- damage manifested by the loss of the original shape of wooden elements (group III) can be considered as insignificant;
- the direction of the relationship is right-hand (positive) for all the 10 tested building elements, but the strength of the correlation between the occurring damage and its technical wear shows a considerable range;
- the technical condition of each of the tested elements also shows the impact of the damage, which is characteristic for the design and material solutions of such an element;
- for the accepted confidence level of 95%, the dependence of moderate strength can be applied to 34-48% of the general population, and the strong correlations to 49-71% of the general population.
In our opinion the results have been thoroughly presented and commented.
New conclusions have been added to the Discussion, section 4 and now it has a doble length in comparison to Results, section 3, so seems to be better balanced. More developed conclusions have already been published and we cannot repat them as to avoid similarity:
- Konior, J. Decision assumptions on building maintenance management. Probabilistic methods, Arch. Civ. Eng., 2007, 53, pp. 403–423.
- Konior, J. Technical Assessment of old buildings by probabilistic approach., Archives of Civil Engineering, 2020, 66(3), pp. 443–466, https://doi.org/10.24425/ace.2020.134407.
- Konior, J. Maintenance of apartment buildings and their measurable deterioration, Tech. Trans. Czas. Tech. 2017, 6, pp. 101–107, https://doi.org/10.4467/2353737xct.17.090.6566.
- Konior, J. Bi-serial correlation of civil engineering building elements under constant technical deterioration, J. Sci. Gen. Tadeusz Kosciuszko Mil. Acad. L. Forces. 2016, 179, pp. 142–150.

Round 2
Reviewer 1 Report
I accept the manuscript. All of my comments are now solved. Explanations regarding the references are appropriate.
Author Response
Wrocław, Poland, 30th April 2021
Dear Reviewer of Materials,
Thank you for the second round review of our paper materials-1189661entitled “Correlation Between Defects and Technical Wear of Materials Used in Traditional Construction” to be published in the journal Materials, Special Issue “Advanced Construction Materials and Processes in Poland”.
We appreciate your thoughtful and accurate comments as well as appreciation of our research works. We have carefully considered all comments and have now completed the revisions incorporating your suggestions in the revised uploaded manuscript.
We hope that the revised paper meets your expectations.
Kind regards,
Jarosław Konior & Mariusz Rejment
Department of Building Engineering, Faculty of Civil Engineering, Wroclaw University of Science and Technology, 50-370 Wrocław, Poland
Here are answers to reviewer’s comments:
REVIEWER 1
General Comments. I accept the manuscript. All of my comments are now solved. Explanations regarding the references are appropriate.
Answer to General Comments. Thank you for appreciation of value of or research works presented in the revised paper. We found your suggestions and comments constructive and valuable which made possible to upgrade our manuscript.

Reviewer 2 Report
The reviewer greatly appreciates the new structure given to the document. The reviewer also agrees with the authors to the extent that they state that their research works “are coherent, consistent, comprehensive, solid and reliable”. However, the reviewer still is of the opinion that the content of the paper does not fully adhere to the Aims & Scope of “Materials”. The paper is undoubtedly about basic building materials (cement, concrete, steel, wood, plaster, brick). Nevertheless, there are many different approaches to the study of materials. The “Aims” Section on the journal website reads: “Our aim is to encourage scientists to publish their experimental and theoretical results in as much detail as possible”
https://www.mdpi.com/journal/materials/about
The emphasis given on experimental and theoretical results leads the reviewer to believe that it is not sufficient to present data abacuses on the state of conservation, however thorough and detailed the data collection and cataloging may be. The abacuses should at least be the starting point of a theoretical analysis, leading to some kind of theoretical model, which frankly cannot be recognized in this paper (statistical data analysis is NOT a theoretical model). Nor is it possible to consider the collection of data on the state of the art as an experimental work in the strict sense.
In conclusion, while acknowledging the value of this work, the reviewer still has several doubts about its correct editorial placement. That said, should the editor be of the opposite opinion, the reviewer does not intend to place further obstacles to publication.
Author Response
Wrocław, Poland, 30th April 2021
Dear Reviewer of Materials,
Thank you for the second round review of our paper materials-1189661entitled “Correlation Between Defects and Technical Wear of Materials Used in Traditional Construction” to be published in the journal Materials, Special Issue “Advanced Construction Materials and Processes in Poland”.
We appreciate your thoughtful and accurate comments as well as appreciation of our research works. We have carefully considered all comments and have now completed the revisions incorporating your suggestions in the revised uploaded manuscript.
We hope that the revised paper meets your expectations.
Kind regards,
Jarosław Konior & Mariusz Rejment
Department of Building Engineering, Faculty of Civil Engineering, Wroclaw University of Science and Technology, 50-370 Wrocław, Poland
Here are answers to reviewer’s comments:
REVIEWER 2
General Comments. The reviewer greatly appreciates the new structure given to the document. The reviewer also agrees with the authors to the extent that they state that their research works “are coherent, consistent, comprehensive, solid and reliable”. However, the reviewer still is of the opinion that the content of the paper does not fully adhere to the Aims & Scope of “Materials”. The paper is undoubtedly about basic building materials (cement, concrete, steel, wood, plaster, brick). Nevertheless, there are many different approaches to the study of materials. The “Aims” Section on the journal website reads: “Our aim is to encourage scientists to publish their experimental and theoretical results in as much detail as possible”
https://www.mdpi.com/journal/materials/about
The emphasis given on experimental and theoretical results leads the reviewer to believe that it is not sufficient to present data abacuses on the state of conservation, however thorough and detailed the data collection and cataloging may be. The abacuses should at least be the starting point of a theoretical analysis, leading to some kind of theoretical model, which frankly cannot be recognized in this paper (statistical data analysis is NOT a theoretical model). Nor is it possible to consider the collection of data on the state of the art as an experimental work in the strict sense.
In conclusion, while acknowledging the value of this work, the reviewer still has several doubts about its correct editorial placement. That said, should the editor be of the opposite opinion, the reviewer does not intend to place further obstacles to publication.
Answer to General Comments. We do appreciate the reviewer’s consistent view on our paper and the subject but cannot agree to full extant.
Firstly, because the paper and the subject represents a special contribution to journal of Materials requested on February 15th 2021 by the editor to submit. Both the title and the abstract were presented to the editor prior to composing the article and submission.
Secondly, because the aim section on the journal website reads: “Materials provides a forum for publishing papers which advance the in-depth understanding of the relationship between the structure, the properties or the functions of all kinds of materials”. Therefore, the relationships = correlation between the structure of defects and the function of the technical wear of basic materials (cement, concrete, steel, timber, plaster, brick) were presented and discussed. Such approach stands for an experimental value of the paper. Indeed, statistical calculation by point biserial coefficient is only a well adopted tool to achieve interesting results of the experiments conducted in situ tenement houses.
Thirdly, because the theoretical grounds regarding intensity of formation of defects in residential buildings and their reliability has already been presented in both papers:
- Konior, J. Intensity of defects in residential buildings and their technical wear, Tech. Trans. Civ. Eng. 2014, 111(2-B), pp. 137–146.
- Konior, J.; Sawicki, M.; Szóstak, M. Intensity of the Formation of Defects in Residential Buildings with Regards to Changes in Their Reliability. Appl. Sci. 2020, 10, 6651. https://doi.org/10.3390/app10196651
The main theoretical assumptions and conclusions: “The concept of the reliability of a residential building is always associated with the performance of exploitation tasks. The performance of a task by a residential building involves its correct fulfilment of certain functions under certain operating conditions and within a specified time. If this function is denoted by j, the building's working conditions by c, and the building's operation time by t, then the task to be performed by the facility can be written as an ordered triple [j,c,t]. By knowing the function that the building needs to perform, it is possible to establish such a set of requirements (wj) for the features of a residential building (characterized by a number of essential and auxiliary technical-operational, economic and other parameters that are important in the process of exploitation and maintenance) that their fulfilment is a necessary and sufficient condition for the correct implementation of the assigned functions (j) by the building. It has been assumed, with some simplifications, that the assessed residential building is, from the point of view of exploitation theory, a two-state object. This means that it may be fit to perform its function (assuming the actual state that is characterized by meeting the requirements (wj)), or unfit to perform its function (assuming the physical state that is characterized by failure to meet the requirements (wj)). The task of the facility, which is then understood as an event (Z) (e.g. with regards to the provision of housing services), is written as the following ordered triple: [wj,c,t]. It was further assumed that the requirements regarding a residential building and its maintenance conditions are known, i.e. the pair [wj,c] is fixed, and consequently it was assumed that the reliability of residential buildings could be assessed as a function of time (t). The adopted model and method of testing a representative group of downtown residential buildings with a traditional construction, which were erected at the turn of the 19th and 20th centuries, indicate that the age of the elements of old residential buildings is of secondary importance in the process of the intensity of loss of their serviceability value. If we assume that the coefficient of determination is the measure of the adjustment of the mathematical models (as a function of the technical wear of building elements over time), which are tested in the nonlinear regression method, then no more than 30% of the element's damage can be explained by the passage of time. Therefore, it is not age that determines the course of the technical wear of the analysed building components. The analysis of the exploitation processes of residential buildings and the transformations of the basic dependencies of the reliability theory indicate that for the service life of an object, in which the time of correct operation to failure has an exponential distribution (it is basically the service life corresponding to the length of operation of the considered residential buildings), the average remaining time of failure-free operation is unchanged at any time. Theoretically, after a certain period of failure-free operation, residential buildings fulfil their functions just like new ones. The optimal moment of renovation occurs after the end of the second period of operation, before the period of rapid wear. Expressing the average duration of the correct failure-free operation of an object by the reliability function R(t), which determines the probability with which the correct operation time of an object will be longer than ti, has a practical application in the exploitation of a residential building and its components. The study of the course of the damage intensity function λ(t) over time reflects the wear process of a residential building in a representative sample of downtown residential buildings erected using traditional methods.”
To sum up, we hope that the above thorough explanation may resolve doubts of the reviewer and proved that our research works are of Materials interest.

Reviewer 3 Report
The authors addressed all the provided comments, except lacking appropriate formatting and page numbering in some parts. I believe these issues can be fixed during copy editing.
Author Response
Wrocław, Poland, 30th April 2021
Dear Reviewer of Materials,
Thank you for the second round review of our paper materials-1189661entitled “Correlation Between Defects and Technical Wear of Materials Used in Traditional Construction” to be published in the journal Materials, Special Issue “Advanced Construction Materials and Processes in Poland”.
We appreciate your thoughtful and accurate comments as well as appreciation of our research works. We have carefully considered all comments and have now completed the revisions incorporating your suggestions in the revised uploaded manuscript.
We hope that the revised paper meets your expectations.
Kind regards,
Jarosław Konior & Mariusz Rejment
Department of Building Engineering, Faculty of Civil Engineering, Wroclaw University of Science and Technology, 50-370 Wrocław, Poland
Here are answers to reviewer’s comments:
REVIEWER 3
General Comments. The authors addressed all the provided comments, except lacking appropriate formatting and page numbering in some parts. I believe these issues can be fixed during copy editing.
Answer to General Comments. Thank you for appreciation of value of or research works presented in the revised paper. We found your suggestions and comments constructive and valuable which made possible to upgrade our manuscript. There is something wrong in page numbering of the template document and definitely must be fixed at the editing stage.

Reviewer 4 Report
Accept in present form.
Author Response
Wrocław, Poland, 30th April 2021
Dear Reviewer of Materials,
Thank you for the second round review of our paper materials-1189661entitled “Correlation Between Defects and Technical Wear of Materials Used in Traditional Construction” to be published in the journal Materials, Special Issue “Advanced Construction Materials and Processes in Poland”.
We appreciate your thoughtful and accurate comments as well as appreciation of our research works. We have carefully considered all comments and have now completed the revisions incorporating your suggestions in the revised uploaded manuscript.
We hope that the revised paper meets your expectations.
Kind regards,
Jarosław Konior & Mariusz Rejment
Department of Building Engineering, Faculty of Civil Engineering, Wroclaw University of Science and Technology, 50-370 Wrocław, Poland
